# Dismantling the Illusion of Vision-Language-Action Models Competence via Explicit Distributional Shifts

Xueyang Zhou [1] [*]   Yangming Xu [1] [*]   Guiyao Tie [1]   Chaoran Hu [1]   Bo Tao [1]   Xingwei Zhao [1]   Xiang Xiang [1]   Pan Zhou [1]   Lichao Sun [2]   Yongchao Chen [3]

## Abstract

Given that simulation can never exhaustively enumerate reality, generalization is the determining factor for whether Vision-Language-Action (VLA) models can translate benchmark success into real-world functionality. However, current evaluation protocols often incentivize mechanical memorization rather than robust policy learning, leading to a paradoxical duality of failure: high-scoring models exhibit *spurious invariance* to semantic changes while simultaneously displaying *extreme brittleness* to trivial environmental perturbations. To address this, we introduce **LIBERO-Gen**, a diagnostic benchmark systematically designed to shift evaluation from intuition-driven heuristics to explicit distributional assumptions. Through a hierarchical protocol spanning **In-distribution**, **Compositional**, and **Domain Generalization**, LIBERO-Gen reveals performance stratifications previously masked by standard metrics. Our analysis identifies Pi0.5 as the top performer (64.0% in Spatial-CG; 21.2% in Task-CG). By identifying perceptual instability and action binding collapse as primary failure modes while validating the efficacy of structured "Stair" sampling, LIBERO-Gen establishes a rigorous baseline for deployment reliability.

## 1. Introduction

Vision–Language–Action (VLA) models have become a central paradigm for embodied agents, unifying perception, understanding, and action  (Sun et al., 2025). Due to the heterogeneity and cost of real-world hardware, simulation benchmarks are indispensable: they provide standardized, reproducible testbeds and effectively steer methodological progress (James et al., 2020; Liu et al., 2023; Mees et al., 2022; Li et al., 2024; Nasiriany et al., 2024). However, we find that existing benchmarks exemplified by LIBERO (Liu et al., 2023) suffer from a fundamental evaluation limitation. Their test protocols largely mirror the training distribution, differing only through visually subtle and often imperceptible state perturbations. Under such settings, high success rates primarily indicate a scale-enabled capacity to fit a narrow task distribution rather than robust policy learning and generalization (Fang et al., 2025b; Zhou et al., 2025b).

As shown in Figure 1, the high performance of existing VLA models on LIBERO obscures a paradoxical duality of failure. On one hand, models exhibit Spurious Invariance: in controlled intervention experiments, such as replacing the target object or corrupting instructions into meaningless tokens, the resulting action trajectories remain nearly unchanged. On the other hand, they simultaneously display Extreme Brittleness: trivial environmental perturbations (e.g., minor visual noise or lighting shifts) can cause catastrophic failure. This contrast reveals a reliance on mechanical memorization rather than causal understanding, severely limiting their adaptability to the physical world.

***Given that simulation can never exhaustively enumerate reality, generalization is not optional—it is the determining factor for whether VLA models can translate from benchmark success to real-world functionality.*** To address this impasse, we introduce LIBERO-Gen. Our central premise is that generalization should not be evaluated through intuition-driven heuristics, but instead grounded in explicit distributional assumptions and diagnostically constructed shifts. For instance, it is unrealistic to expect a model trained under a single context and a fixed set of distractors to spontaneously acquire immunity to distribution shift. To this end, LIBERO-Gen employs a rigorous, theory-guided hierarchical evaluation, systematically probing generalization boundaries across five key dimensions: background, language, distractors, task semantics, and spatial configuration. This evaluation is structured into three

---
[*]Equal contribution   [1] Huazhong University of Science and Technology, Wuhan, China   [2] Lehigh University, Bethlehem, PA, USA   [3] College of AI, Tsinghua University, Beijing, China .   Correspondence to: Pan Zhou <panzhou@hust.edu.cn>, Bo Tao <taobo@hust.edu.cn>, Yongchao Chen <yongchaochen12@gmail.com>.

*Proceedings of the 43rd International Conference on Machine Learning*, Seoul, South Korea. PMLR 306, 2026. Copyright 2026 by the author(s).

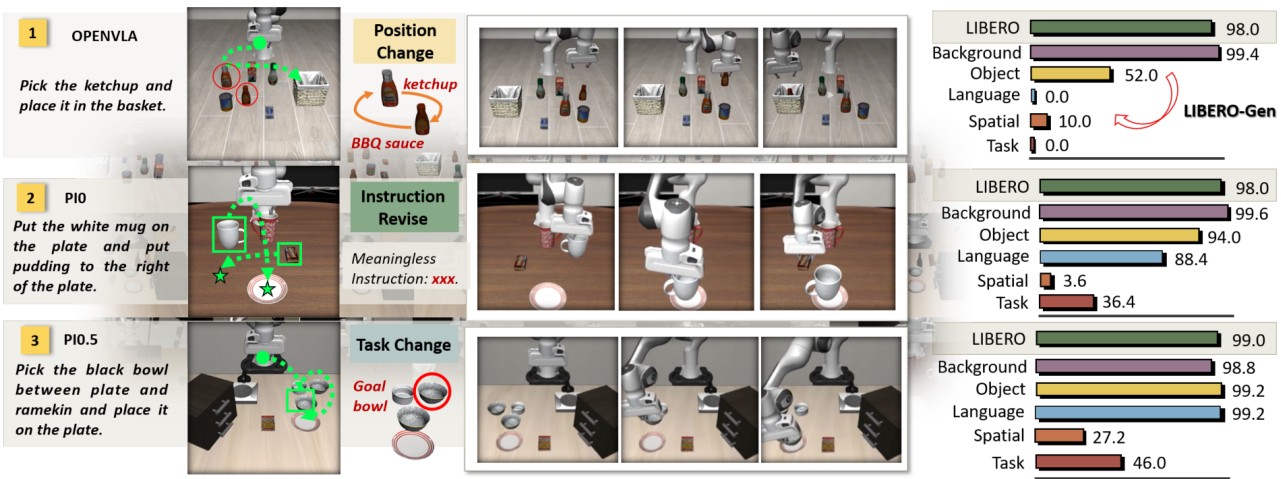

*Figure 1.* Behavior of OpenVLA, Pi0, and Pi0.5 under spatial and semantic perturbations. Despite 100% success on the Original Task, models exhibit rigid behaviors in modified settings: (1) Position Change: Swapping object positions leads models to grasp the distractor at the original coordinates; (2) Task Change & Meaningless Instruction: Altering the text to a new target or random characters results in models persisting with the original action sequence.

distinct levels: (1) **In-distribution.** Verifies the foundational capability to fit the training support set. (2) **Compositional Generalization.** (Gao et al., 2024) Tests adaptability to novel combinations of known factors. (3) **Domain Generalization.** (Besserve et al., 2021) Probes robustness against completely unseen distribution shifts (e.g., unseen textures). Our key contributions are summarized as follows:

- We demonstrate that current protocols are insufficient to support the simulation-to-reality transition, emphasizing the critical necessity of robust generalization for ensuring reliable VLA deployment.

- We introduce a hierarchical evaluation suite LIBERO-Gen that shifts the paradigm from intuition-driven heuristics to explicit distributional assumptions, employing "staircase" sampling to systematically probe generalization boundaries.

- While identifying Pi0.5 as the top performer, we pinpoint perceptual instability and action binding collapse as primary failure modes, providing a rigorous baseline and direction for reliable VLA deployment.

## 2. Formalization

**Vision-Language-Action Model.** We formalize the VLA model as a vision-language-conditioned generative policy $\pi_\theta(a_t|o_t, l)$ within a partially observable Markov Decision Process (POMDP) framework for task execution. The goal of VLA is to model the conditional distribution of actions $P(a_t \mid o_t, l)$, where $o_t$ is the visual observation and $l$ is the linguistic instruction at time $t$.

For a given task instance, the environment is initialized from a state distribution $s_0 \sim \rho(\cdot)$. At time $t$, the model receives an observation $o_t \in \mathcal{O}$ and a natural-language instruction $l \in \mathcal{L}$, generating an action $a_t \in \mathcal{A}$. The environment dynamics and observation function are defined by:

$$s_{t+1} \sim P(\cdot \mid s_t, a_t), \quad o_t = \Omega(s_t). \tag{1}$$

Task completion is evaluated using a binary success indicator $\mathrm{Succ}(\tau) \in \{0, 1\}$, where $\tau = \{(o_t, a_t)\}_{t=0}^{T-1}$ is the episode rollout trajectory of horizon $T$. In common VLA paradigms (e.g., LIBERO), $\mathrm{Succ}(\tau)$ is typically determined by an environment-defined task verifier checking for specific goal criteria (e.g., object pose/contact constraints).

**Compositional Generalization.** We formalize compositional generalization for VLA policies by decomposing each task instance into causal factors. A task is modeled as a tuple:

$$x = (\tau, c_1, c_2, \ldots, c_n), \tag{2}$$

where $\tau \in \mathcal{T}$ denotes the abstract task semantics (e.g., pick-and-place, push, stack), and $c_i \in C_i$ denotes independent context factors such as visual appearance, instruction phrasing, and spatial layouts. Let $\mathcal{C} = \mathcal{C}_1 \times \cdots \times \mathcal{C}_n$ denote the joint context space. The training dataset $\mathcal{D}_{\text{train}}$ is sampled from a **restricted support** of the full Cartesian product:

$$\mathcal{D}_{\text{train}} \subsetneq \mathcal{T} \times \mathcal{C}. \tag{3}$$

Many factor combinations may be absent, even though individual factor values are observed during training:

$$\exists (\tau, c) \in \mathcal{T} \times \mathcal{C} \quad \text{such that} \quad \begin{aligned} &(\tau, c) \notin \mathcal{D}_{\text{train}}, \\ &\tau \in \text{proj}_{\mathcal{T}}(\mathcal{D}_{\text{train}}), \\ &c_i \in \text{proj}_{\mathcal{C}_i}(\mathcal{D}_{\text{train}}). \end{aligned} \tag{4}$$

A VLA policy $\pi_\theta$ demonstrates compositional generaliza-

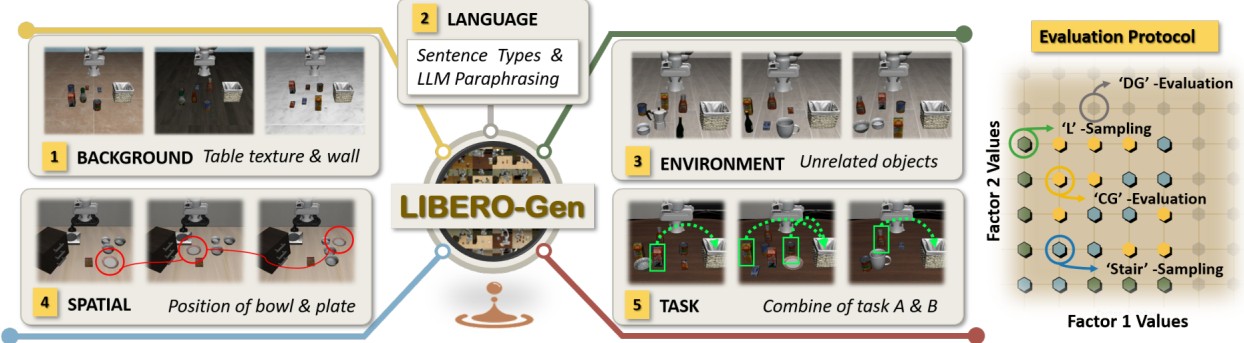

*Figure 2.* Overview of the LIBERO-Gen evaluation framework. The right panel illustrates the data sampling strategies in the factor space. For training, we employ 'L' (green points) and 'Stair' (blue points) configurations. For evaluation, the framework specifically tests CG (yellow points) and DG (grey points) to assess model robustness across different distribution shifts.

tion if, given a training set $\mathcal{D}_{\text{train}}$ with limited compositional coverage and restricted factor combinations, it achieves low execution error on task-context pairs composed of novel combinations of seen factors without extra supervision. Let $\mathcal{D}_{\text{eval}}^{(1)}$ be an evaluation distribution such that:

$$\mathcal{D}_{\text{eval}}^{(1)} \cap \mathcal{D}_{\text{train}} = \varnothing,$$

$$\text{proj}_{\mathcal{T}}\left(\mathcal{D}_{\text{eval}}^{(1)}\right) \subseteq \text{proj}_{\mathcal{T}}(\mathcal{D}_{\text{train}}), \qquad (5)$$

$$\text{proj}_{\mathcal{C}_i}\left(\mathcal{D}_{\text{eval}}^{(1)}\right) \subseteq \text{proj}_{\mathcal{C}_i}(\mathcal{D}_{\text{train}}), \quad \forall i.$$

Then, $\pi_\theta$ exhibits compositional generalization if:

$$\mathbb{E}_{(\tau,c)\sim\mathcal{D}_{\text{eval}}^{(1)}} \left[\mathbb{E}_{\tau\sim\pi_\theta(\cdot|\tau,c)} \left[1 - \text{Succ}(\tau)\right]\right] \leq \epsilon, \qquad (6)$$

for a small constant $\epsilon > 0$.

**Domain Generalization.** We next formalize domain generalization (DG) for VLA policies, which captures robustness to *out-of-support* environmental shifts that cannot be expressed as recombinations of training factors. We assume that task execution is governed by a set of latent, conditionally independent generative mechanisms $\{M_j\}_{j=1}^{J}$ (e.g., task semantics, object dynamics, language grounding, visual appearance), following the Independent Mechanisms principle (Besserve et al., 2021). A *domain* $e \in \mathcal{E}$ is defined as a particular instantiation of a subset of these mechanisms, inducing an environment-specific observation distribution:

$$(o_t, l) \sim P_e(o, l \mid s_t), \qquad (7)$$

while sharing the same underlying task semantics $\tau$ and transition dynamics $P(s_{t+1} \mid s_t, a_t)$.

Let $\mathcal{E}_{\text{train}} \subsetneq \mathcal{E}_{\text{all}}$ denote the set of domains available during training, and let $\mathcal{E}_{\text{eval}} \subset \mathcal{E}_{\text{all}} \setminus \mathcal{E}_{\text{train}}$ denote unseen test domains, and $c \in \mathcal{C}$ represents context variables instantiated

under domain $e$. The training dataset is sampled as:

$$\mathcal{D}_{\text{train}} \sim \bigcup_{e \in \mathcal{E}_{\text{train}}} P_e(\tau, c). \qquad (8)$$

Unlike compositional generalization, domain generalization allows evaluation samples whose context realizations lie *outside* the convex hull of training support:

$$
\begin{aligned}
& e \notin \mathcal{E}_{\text{train}}, \\
\exists(\tau, c, e) \quad \text{s.t.} \quad & c \notin \text{supp}(\mathcal{D}_{\text{train}}), \\
& \tau \in \text{proj}_{\mathcal{T}}(\mathcal{D}_{\text{train}}).
\end{aligned}
\qquad (9)
$$

A VLA policy $\pi_\theta$ exhibits *domain generalization* if it achieves low execution error when deployed in unseen domains whose underlying generative mechanisms differ from those observed and learned during training, without access to extra domain labels or adaptation data.

Formally, $\pi_\theta$ is domain-generalizable if:

$$\mathbb{E}_{e\sim\mathcal{E}_{\text{eval}}} \left[\mathbb{E}_{(\tau,c)\sim P_e} \left[\mathbb{E}_{\tau\sim\pi_\theta(\cdot|\tau,c,e)} \left[1 - \text{Succ}(\tau)\right]\right]\right] \leq \epsilon,$$
$$(10)$$

for a small constant $\epsilon > 0$.

## 3. LIBERO-Gen: Benchmark Design

We instantiate the LIBERO-Gen benchmark across five distinct dimensions, as shown in Figure 2. For each dimension, we strictly define the Training Protocol (designed to induce spurious correlations) and the Evaluation Protocol (designed to test generalization bounds).

### 3.1. Hierarchical Generalization Design

Building on the factorized formulation in Sec. 2, we characterize the generalization boundary of VLA policies under *controlled* distribution shifts by selecting a set of manipulable factors and systematically intervening on them at evaluation time. Each task instance is specified by an abstract task

semantic $\tau \in \mathcal{T}$ together with a context configuration $c \in \mathcal{C}$, where $c$ denotes the joint assignment of the factor subset considered in this suite. The training set $\mathcal{D}_{\text{train}} \subset \mathcal{T} \times \mathcal{C}$ is sampled from a restricted support over task–context pairs, so that certain portion of valid combinations in $\mathcal{T} \times \mathcal{C}$ are never observed during the policy training process.

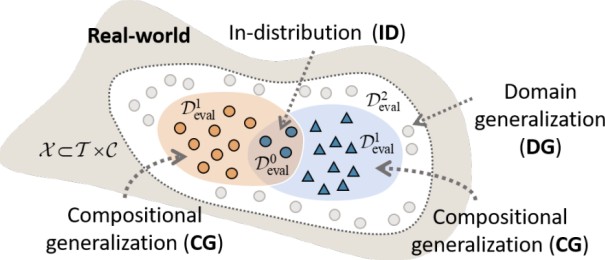

*Figure 3.* Sampling Distributions for Training and Evaluation.

As shown in Figure 3, we evaluate a policy $\pi_\theta$ on a suite of datasets $\{\mathcal{D}^k_{\text{eval}}\}_{k \in \{0,1,2\}}$, where $k$ explicitly corresponds to an increasing order of distribution shift severity, ranging from identical training support to entirely novel domains:

- **In-distribution (ID) evaluation** ($k = 0$). $\mathcal{D}^0_{\text{eval}}$ contains task–context pairs drawn from the same empirical support as $\mathcal{D}_{\text{train}}$ to assess policy under non-shifted conditions.

- **Compositional generalization** ($k = 1$). $\mathcal{D}^1_{\text{eval}}$ consists of *unseen* task–context pairs $(\tau, c) \notin \mathcal{D}_{\text{train}}$ formed by recombining *previously observed* tasks and contexts. Concretely, it satisfies $\mathcal{D}^1_{\text{eval}} \cap \mathcal{D}_{\text{train}} = \varnothing$, $\text{proj}_\mathcal{T}(\mathcal{D}^1_{\text{eval}}) \subseteq \text{proj}_\mathcal{T}(\mathcal{D}_{\text{train}})$, and $\text{proj}_\mathcal{C}(\mathcal{D}^1_{\text{eval}}) \subseteq \text{proj}_\mathcal{C}(\mathcal{D}_{\text{train}})$.

- **Domain generalization (DG)** ($k = 2$). $\mathcal{D}^2_{\text{eval}}$ evaluates robustness to *unseen* appearance domains outside the training support. We treat each texture/material set as a domain and adopt the standard DG protocol that trains on a set of available domains $E_{\text{avail}}$ and tests on a larger domain set $E_{\text{all}} \supset E_{\text{avail}}$, thereby probing genuine out-of-distribution shifts beyond factor recombination.

### 3.2. Multi-dimensional Evaluation

Our suite measures robustness along five complementary dimensions: (i) **background** (textures/materials of the floor, walls, and table), (ii) **language** (instruction wording and paraphrases), (iii) **distractor objects** (irrelevant objects present in the scene), (iv) **task** (the task semantic goal), and (v) **spatial configuration** (object and target positions/layouts). For each dimension, we specify the training distribution and evaluation sampling strategy as follows.

**Visual Generation.** To instantiate the ***background*** dimension on the LIBERO-Object benchmark, we define the two independent variables as the task semantics ($\tau \in \mathcal{T}$) and the tabletop textures ($c \in \mathcal{C}$), constructing the training set as a subset of their compositional pairs. For the training

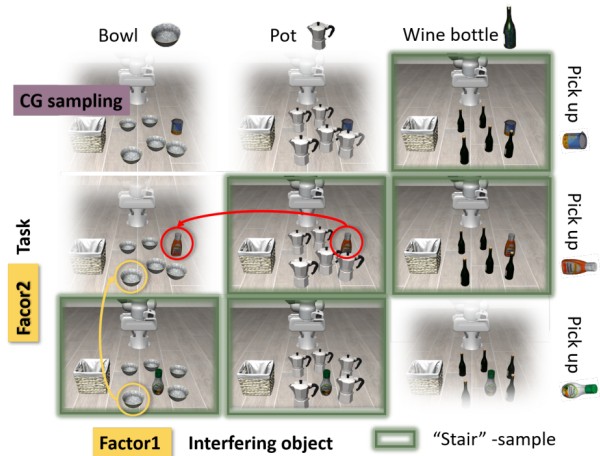

*Figure 4.* Example of Sampling for Distractor Generation.

distribution, we select five tabletop textures as the seen set $\mathcal{C}_{\text{seen}} = \{c_1, \ldots, c_5\}$, partition the 10 LIBERO-Object base tasks into five disjoint groups with two tasks per group, and use a structured task–texture split in which each group is paired with a single texture (see Appendix C.1 for detailed texture assets and training configurations in Table 6). At test time, we evaluate the same set of tasks under three levels of controlled texture perturbations (refer to Table 7 in Appendix C.1).

**Language Generation.** To instantiate the ***language*** dimension on the LIBERO-Object benchmark, we define the two independent variables as the task semantics ($\tau \in \mathcal{T}$) and the linguistic templates ($c \in \mathcal{C}$), constructing the training set as a subset of their compositional pairs. The training distribution uses five semantically invariant but syntactically distinct instruction templates as $\mathcal{C}_{\text{seen}} = \{c_1, \ldots, c_5\}$, partitions the 10 tasks into five groups following the same division rule, and pairs each task group with a single instruction template to form a structured task–language split (detailed training configurations are provided in Table 10 of Appendix C.3). For evaluation sampling, we set three generalization levels (specific evaluation templates and paraphrase examples are listed in Table 11 of Appendix C.3).

**Distractor Generation.** To instantiate the ***distractor*** dimension on the LIBERO-Object benchmark, we define the two independent variables as the task semantics ($\tau \in \mathcal{T}$) and the distractor configurations ($c \in \mathcal{C}$), constructing the training set as a subset of their compositional pairs. We construct five distractor configurations as $\mathcal{C}_{\text{seen}} = \{c_1, \ldots, c_5\}$ by selecting distinct irrelevant object categories with multiple instances per category, and pair each task group with a corresponding distractor configuration to obtain a structured task–distractor split (see Appendix C.2 for detailed distractor assets and training settings in Table 8). During evaluation, we set three generalization levels (fixed evaluation configurations are shown in Table 9 of Appendix C.2).

*Table 1.* Evaluation Results on LIBERO-Gen Benchmark. Success rates (%) across different generalization axes: Original (Ori.), In-Distribution ($k1$), Compositional Generalization ($k2$), and Domain Generalization ($k3$). **Bold** results indicate the best performance, and underlined results indicate the worst performance among base models.

| Models | Background | | | | Language | | | | Unrelated Object | | | | Position | | | | Task | | | |
|---|---|---|---|---|---|---|---|---|---|---|---|---|---|---|---|---|---|---|---|---|
| | Ori. | $k1$ | $k2$ | $k3$ | Ori. | $k1$ | $k2$ | $k3$ | Ori. | $k1$ | $k2$ | $k3$ | Ori. | $k1$ | $k2$ | $k3$ | Ori. | $k1$ | $k2$ | $k3$ |
| OpenVLA | 98.0 | **99.6** | 94.0 | 99.4 | 98.0 | 0.0 | 0.0 | 39.2 | 98.0 | 67.2 | 63.8 | 52.0 | 98.0 | 51.5 | 6.8 | 10.0 | 93.0 | 95.0 | 0.0 | 0.0 |
| Pi0 | 98.0 | 97.6 | 95.6 | **99.6** | 98.0 | 96.8 | **99.2** | 94.0 | 98.0 | 90.4 | 95.2 | 88.4 | 95.4 | 93.2 | 7.2 | 3.6 | 95.0 | 98.8 | 13.6 | 36.4 |
| Pi0.5 | **99.0** | 99.2 | **99.0** | 98.8 | **99.0** | **98.8** | 98.8 | **99.2** | **99.0** | **99.2** | **99.2** | **99.2** | 98.8 | **97.2** | 49.2 | **27.2** | 96.4 | **99.6** | **21.2** | **46.0** |
| Molmoact | 95.4 | 94.0 | 92.0 | 88.0 | 95.4 | 86.0 | 84.0 | 90.0 | 95.4 | 90.0 | 90.0 | 88.0 | 87.0 | 87.6 | 2.8 | 0.8 | 92.0 | 87.6 | 6.0 | 6.0 |
| NORA | 95.4 | 82.4 | 94.0 | 93.2 | 95.4 | 93.4 | 93.2 | 93.2 | 95.4 | 70.8 | 70.0 | 0.0 | 92.2 | 92.8 | 1.6 | 0.8 | 88.8 | 88.0 | 8.8 | 12.8 |
| x-VLA | 98.6 | 98.0 | 96.8 | 98.4 | 98.6 | 96.8 | 99.0 | 97.0 | 98.6 | 64.8 | 46.0 | 63.8 | 98.2 | 96.2 | 0.0 | 12.8 | **98.1** | 98.9 | 15.6 | 8.2 |
| Univla | 96.8 | 94.0 | 93.0 | 96.0 | 96.8 | 93.0 | 95.0 | 91.0 | 96.8 | 74.0 | 61.0 | 54.0 | 96.5 | 49.0 | 0.0 | 0.0 | 94.4 | 86.2 | 9.0 | 9.0 |
| *Fine-tuned* | | | | | | | | *Models fine-tuned on LIBERO-Gen specific subsets* | | | | | | | | | | | | |
| Pi0* | – | 97.8 | 97.2 | 98.8 | – | 97.6 | **99.2** | 96.0 | – | 95.6 | 98.0 | 97.0 | – | 95.4 | 13.2 | 4.2 | – | 98.8 | 13.6 | 36.4 |
| Pi0.5* | – | 98.0 | 97.0 | 98.4 | – | 97.0 | 98.4 | 97.0 | – | 95.4 | 97.4 | 97.4 | – | **97.2** | **64.0** | 27.0 | – | **99.6** | **21.2** | **46.0** |

**Task Compositionality: Action–Object Generalization.** We instantiate the *task semantic* dimension on LIBERO-Object and LIBERO-10 to evaluate whether VLA policies can generalize across action–object semantics beyond the combinations observed during training. The training data combines 10 single-object manipulation tasks from LIBERO-Object and 3 composite tasks from LIBERO-10, covering all single objects but only three specific object pairs (refer to Appendix C.5 for the full training task list in Table 12). For evaluation sampling, we set three generalization levels (specific evaluation task pairs are detailed in Table 13 of Appendix C.5).

**Spatial Generalization: Layout Extrapolation.** We instantiate the *spatial* dimension on LIBERO-Spatial, where each task relocates a bowl initialized at one of 10 discrete locations to a plate. We take the bowl and plate initial locations as the context variables, i.e., $c = (p_{bowl}, p_{plate})$. For the training distribution, we construct a structured spatial split over the bowl–plate location grid by including the original 10 tasks that move the bowl from each location to the fixed plate, together with 9 auxiliary tasks where the bowl is fixed at $p_{bowl}^\star$ and the plate is placed at the other 9 locations (see Appendix C.4 for detailed spatial assets and sampling protocols). At test time, we set three generalization levels (configurations are detailed in Appendix C.4).

### 3.3. Evaluation Protocol

Based on the five evaluation suites described above and the three perturbation regimes, we design a unified protocol to assess the generalization of each VLA policy.

**Suites and perturbation regimes.** Let $\mathcal{Q} = \{\text{Background, Language, Distractors, Task, Spatial}\}$ denote the set of evaluation suites (one per dimension) and $\mathcal{K} = \{0, 1, 2\}$ denote the perturbation regimes (ID, compositional generalization, and OOD/DG, respectively). For each suite

$q \in \mathcal{Q}$ and regime $k \in \mathcal{K}$, we evaluate each task $t \in \mathcal{T}_q$ by rolling out the policy from $N$ independently sampled initializations drawn from the suite-specific distribution $\mathcal{D}_{eval}^{q,k}$.

**Task-level success.** We report the task success rate as

$$S_{q,k}(t) = \frac{1}{N} \sum_{i=1}^{N} \text{Succ}\left(\tau_{q,k,i}^{(t)}\right), \qquad (11)$$

where $\tau_{q,k,i}^{(t)}$ denotes the complete executed trajectory in the $i$-th rollout. Correspondingly, $(\mathcal{E}_{q,k,i}^{(t)}, l_{q,k,i}^{(t)})$ represent the environment initialization (including the suite-specific perturbation) and language instruction provided to the policy. Here, the binary indicator function $\text{Succ}(\tau) \in \{0, 1\}$ indicates whether the rollout successfully completes task $t$.

## 4. Experiments

In this section, we present a comprehensive empirical evaluation on LIBERO-Gen. Our experiments address three pivotal research questions:

- **RQ1:** Do high simulation scores truly reflect the robustness required for real-world functionality?

- **RQ2:** What specific failure modes and generalization challenges hinder VLA adaptation to distribution shifts?

- **RQ3:** How can models trained on finite data achieve the robust generalization needed for reliable deployment?

### 4.1. Main Results

We conduct a comprehensive evaluation of representative VLA models on LIBERO-Gen, with detailed results summarized in Table 1.

(1) **The Illusion of Competence.** While most models saturate standard benchmarks with >95% success rates, LIBERO-Gen reveals this performance stems from mechan-

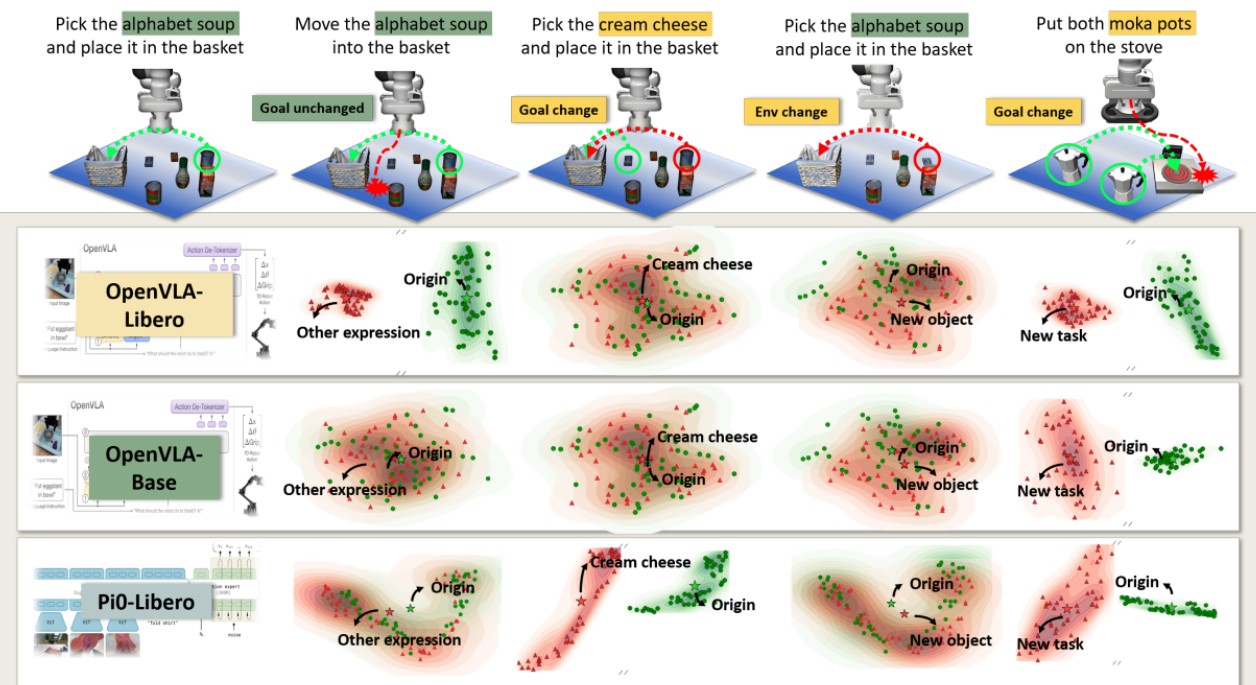

*Figure 5.* Latent Feature Visualization under Distribution Shifts.

ical memorization rather than robust understanding. For instance, OpenVLA matches Pi0.5 on original tasks but suffers catastrophic collapse under minimal variations. Specifically, it drops to 0.0% under semantic paraphrasing and 67.2% in the presence of irrelevant clutter, exposing a fundamental brittleness to minor, goal-irrelevant perturbations.

(2) **Variable-Dependent Generalization Hierarchy.** Generalization difficulty varies significantly across distinct types of distribution perturbations. Models exhibit high robustness to Background variations (mostly >90% success). However, Position and Task generalization emerge as the primary bottlenecks; nearly all baselines (e.g., OpenVLA, Molmoact, Univla) plummet to near-zero success rates in these categories. This distinct hierarchy highlights that current VLA architectures struggle most with spatial extrapolation and compositional task planning.

(3) **Differentiation via Hierarchical Generalization.** LIBERO-Gen effectively stratifies capabilities that standard metrics conflate. While OpenVLA and x-VLA fail to generalize on spatial tasks (achieving 6.8% and 0.0% on Position-CG, respectively), Pi0.5 demonstrates superior structural generalization, achieving 49.2% on Position-CG and 46.0% on Task-DG. Furthermore, models like NORA and x-VLA show competitive semantic understanding on Language tasks (>93%) but lack the physical grounding required for spatial robustness.

(4) **Decoupling Model Capabilities from Data Bias.** The

prevailing generalization failures should not be ascribed solely to architectural deficits; rather, they reflect the inherent impossibility of deriving unbiased policies from biased training distributions. LIBERO-Gen addresses this by establishing a rigorous protocol that ensures fair capability assessment under controlled distributions. This value is empirically validated by our fine-tuning results: when trained under our structured regime, models demonstrate significantly unlocked potential, with Pi0 and Pi0.5 improving their Position-CG performance from 7.2% to 13.2% and 49.2% to 64.0%, respectively.

### 4.2. Failure Mechanisms

To answer **RQ2**, we visualize the latent feature of the encoder under different perturbation regimes. We identify two distinct failure mechanisms:

**Type I: Perceptual Instability.** The encoder fails to marginalize out goal-irrelevant nuisance variables. As illustrated in Figure 5 (Column 2), *OpenVLA-Libero* (Row 1) projects semantically equivalent instructions onto disjoint latent manifolds, directly precipitating task failure. In contrast, *OpenVLA-Base* (Row 2) maintains high feature overlap under similar perturbations. This divergence indicates that distribution biases within the *Libero-object* dataset drive *OpenVLA-Libero* to learn spurious language-vision correlations rather than robust semantic representations.

**Type II: Spurious Invariance vs. Binding Collapse.** Un-

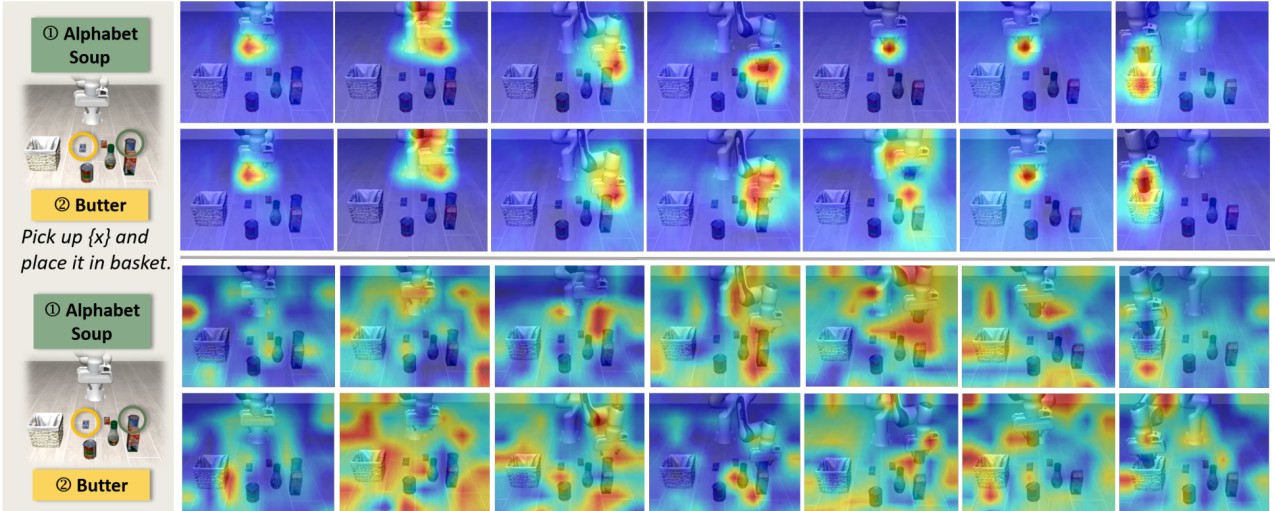

*Figure 6.* Visualizing Attention Inertia and Cross-Modal Disconnect: OpenVLA (Top) vs. Pi0 (Bottom). We observe that both models fail to effectively redirect attention when the instruction shifts from Soup" to Butter".

der task-critical goal shifts, we distinguish two bottlenecks:

- **Spurious Invariance (Encoder Failure):** In object replacement (e.g., *Soup → Cheese*, Column 3-4), all models exhibit significant latent overlap with the original states. This implies the encoders remain invariant to critical visual cues, causing the policy to erroneously persist with the original action plan.

- **Binding Collapse (Representation or Decoder Failure):** Conversely, for "New Tasks" (Column 5), *OpenVLA-Libero* can project OOD states onto distinct latent manifolds. However, latent separability does not guarantee correct action generation. Such failures may stem from either encoder-side representation failure, where the fused representation does not preserve task-critical object separability, or decoder-side binding failure, where the decoder cannot map an unseen task composition to valid actions. For example, in *LIBERO-GOAL*, $\pi_0$ can solve seen tasks such as placing the bowl on the stove and placing the bowl on the plate, but fails on the unseen composition of placing the plate on the stove.

### 4.3. Path to Generalization

To answer **RQ3**, we move beyond black-box optimization and adopt a divide-and-conquer strategy that decouples the vision-language encoder from the policy decoder, applying targeted structural interventions to each component.

**Regularizing Representation Space ($f_{\theta_f}$): Balancing Invariance and Sensitivity.** To mitigate Type I instability and Type II spurious invariance, the encoder must effectively marginalize nuisance variables without sacrificing task-relevant sensitivity.

- **Consistency Regularization via Augmentation.** We in-

*Table 2.* Ablation study results. We compare the OpenVLA baseline (**Origin**) against the model fine-tuned on LIBERO-Gen (**Fine-tuning**) and training with data augmentation (**Enhanced**).

| Regime | Origin | Fine-tuning | | Enhanced | |
|---|---|---|---|---|---|
| | Acc. | Acc. | Δ | Acc. | Δ |
| *Dimension: Background* | | | | | |
| $k_1$ (ID) | 99.6 | 98.8 | -0.8 | **99.6** | 0.0 |
| $k_2$ (CG) | 94.0 | **99.6** | +5.6 | 98.8 | +4.8 |
| $k_3$ (DG) | 99.4 | **99.6** | +0.2 | 98.8 | -0.6 |
| *Dimension: Language* | | | | | |
| $k_1$ (ID) | 0.0 | 91.4 | +91.4 | **98.5** | +98.5 |
| $k_2$ (CG) | 0.0 | 90.2 | +90.2 | **99.2** | +99.2 |
| $k_3$ (DG) | 39.2 | 88.0 | +48.8 | **98.6** | +59.4 |
| *Dimension: Object* | | | | | |
| $k_1$ (ID) | 67.2 | **78.4** | +11.2 | 86.2 | +19.0 |
| $k_2$ (CG) | 63.8 | **81.8** | +18.0 | 88.0 | +24.2 |
| $k_3$ (DG) | 52.0 | **66.0** | +14.0 | 78.4 | +26.4 |

troduce a modality-specific consistency loss to enforce logical constraints on the latent manifold. Formally, we optimize for invariance against nuisance perturbations $\text{aug}_{\text{nuis}}$ (i.e., $f_{\theta_f}(\text{aug}_{\text{nuis}}(x)) \approx f_{\theta_f}(x)$) while maximizing divergence under critical state shifts $\delta_{\text{crit}}$ (i.e., $\|f_{\theta_f}(x + \delta_{\text{crit}}) - f_{\theta_f}(x)\| > \epsilon$). As shown in Table 2, this explicitly decouples visual noise from semantic content, significantly enhancing robustness against background and linguistic distractions without requiring additional action labels.

- **Architectural Insight: Cross-Modal Filtering.** Embodied control demands strict instruction-conditional filtering. However, Figure 6 reveals that VLA exhibits "attention inertia": altering the target (e.g., *Soup → Cheese*) fails to redirect visual focus. We attribute this to the naive to-

ken concatenation used in baselines, which lacks explicit feature interaction. To structurally prevent such spurious correlations, we advocate for dense *cross-modal attention* mechanisms to dynamically re-weight visual patches conditional on linguistic intent.

We further corroborate this diagnosis through a **linear probing analysis** on the model's fused hidden states, as show in Figure 7. By training classifiers to predict the target object specified by the instruction, we uncover a catastrophic *binding collapse*: despite balanced evaluation samples, the model yields **0.00% accuracy** on specific object categories (e.g., *Milk*, *Butter*), while consistently misclassifying diverse inputs as high-saliency objects (e.g., *Cookies*, *Ketchup*). This "mode collapse" in the feature space confirms that naive token concatenation fails to suppress task-irrelevant visual signals.

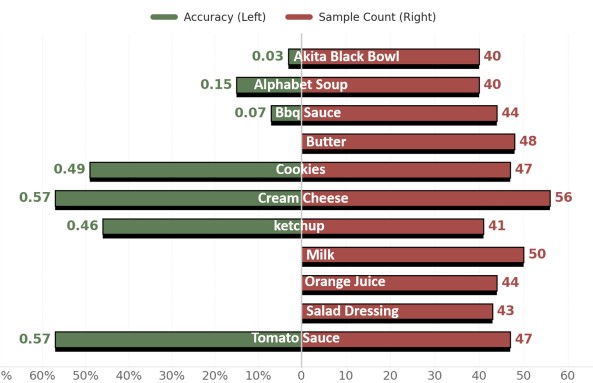

Figure 7. Linear Probing Analysis of Cross-Modal Binding. The chart displays the recognition accuracy (left, green) versus the sample count (right, red) for each object category.

**Enhancing Action Binding.** Generalization in the decoder requires robustly binding novel features to valid action sequences. We address this through improved data topology and output space formulation.

- **Structured Data Sampling:** Addressing the intractability of covering real-world diversity, we conducted a controlled comparison between random sampling and our structured "Stair" sampling under identical data budgets. As illustrated in Figure 8, the structured approach yields significant gains over random sampling. For example, pi0.5 improves from 52.3% to 64.0% in Position generalization and from 14.2% to 21.2% in Task generalization. This demonstrates that the topological structure of "Stair" sampling creates a more uniform variable exposure, preventing the decoder from overfitting to sampling biases and thereby facilitating the learning of generalized control rules rather than memorizing dominant modes.

- **Continuous vs. Discrete Action Modeling.** Empirical results highlight the autoregressive **OpenVLA** fails completely on Task generalization (**0.0%**), exhibiting se-

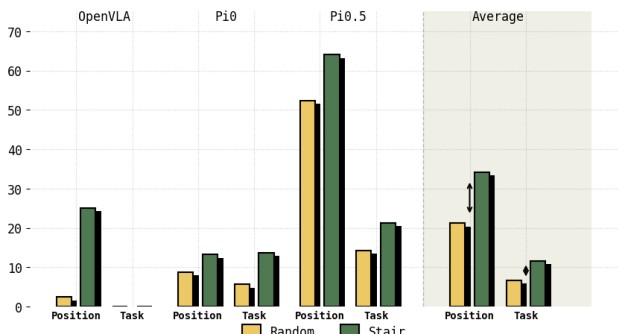

Figure 8. Generalization Comparison under Different Sampling Strategies. We compare the performance of models trained using **Random** (yellow) versus structured **Stair** (green) sampling strategies across Position and Task generalization.

vere "Binding Collapse." In contrast, the continuous flow-matching **Pi0.5** retains significantly higher robustness. While acknowledging that disparate model capacities and pre-training data confound a strictly controlled comparison, these results suggest that continuous trajectory modeling enables smoother interpolation and superior adaptation compared to rigid discrete binning.

## 5. Related Work

**Vision-Language-Action Models.** Vision-language-action (VLA) models have emerged as a unifying paradigm for embodied intelligence, enabling robotic agents to map visual observations and language instructions into executable actions (Kim et al., 2024; Black et al., 2024; Bu et al., 2025; Qu et al., 2025; Cen et al., 2025; Li et al., 2025; Liu et al., 2025; Sun et al., 2025; Deng et al., 2025; Ma et al., 2025). Beyond individual policy architectures, recent studies also explore agentic robot frameworks, embodied AI development workflows, and broader language-model-based robotics systems (Yang et al., 2025b; Zhou et al., 2026; Zeng et al., 2023). In parallel, post-training techniques have become increasingly important for adapting large foundation models to downstream tasks (Tie et al., 2025). Despite this rapid progress, the field still faces a critical bottleneck: current evaluation protocols often fail to predict real-world reliability (Yang et al., 2025a; Sun et al., 2024; Zhang et al., 2025b; Jain et al., 2025). Bridging the gap between simulation performance and physical deployment therefore requires standardized benchmarks that can diagnose not only whether a model succeeds, but also under what type of distributional shift it fails.

**Benchmarks for VLA Evaluation.** To support systematic evaluation, various benchmarks (Fan et al., 2026; Guruprasad et al., 2024; Sedlacek et al., 2025; Mees et al., 2022; Walke et al., 2023) have been proposed, including RL-Bench (James et al., 2020), VLABench (Zhang et al., 2025c), RoboTwin (Mu et al., 2024), and VLA-Arena (Zhang et al.,

2025a). Among them, LIBERO (Liu et al., 2023) has established itself as a de facto standard for VLA evaluation. Its standardized task suites and unified reporting metrics have enabled fair comparison across models and have made it a primary benchmark in recent VLA research.

Recent benchmarks have further extended VLA evaluation toward robustness, out-of-distribution behavior, and broader generalization. LIBERO-Plus (Fei et al., 2025) focuses on perturbation robustness by introducing heuristic variations to existing tasks. THE COLOSSEUM (Pumacay et al., 2024) evaluates manipulation robustness under environmental and physical changes. AGNOSTOS (Zhou et al., 2025a) studies cross-task generalization by testing policies on unseen tasks, while INT-ACT (Fang et al., 2025a) examines the gap between intention understanding and action execution under unseen objects and language variations. RoboTwin (Mu et al., 2024), as a domain-randomization-related benchmark, evaluates policy robustness under broad unseen variations in objects, layouts, and environments. These benchmarks are valuable for exposing model fragility from different perspectives, but their notions of OOD and generalization are often defined through heuristic unseen cases, such as unseen perturbations, unseen tasks, unseen objects, unseen language instructions, or environmental changes.

**Robustness and Generalization Gaps.** While robustness-oriented benchmarks such as LIBERO-Pro (Zhou et al., 2025d) and VLA-Arena (Zhang et al., 2025a) have advanced VLA evaluation by introducing environmental perturbations, distractors, and real-world variations, they typically treat training-unseen cases as a broad OOD category. Related security studies further reveal that multimodal and VLA systems can be vulnerable to adversarial or backdoor attacks (Zhou et al., 2025c; Cai et al., 2026). Such evaluations and analyses are effective for exposing model fragility, but they do not explicitly characterize the distributional structure behind different generalization failures.

LIBERO-Gen complements these benchmarks by evaluating VLA generalization under controlled data-distribution assumptions. Instead of treating all unseen cases uniformly, it distinguishes in-distribution generalization (ID), compositional generalization (CG), and domain generalization (DG), enabling a more fine-grained diagnosis of when and why VLA models fail to generalize.

## 6. Conclusion

We challenge the "illusion of competence" in current VLA evaluation by introducing LIBERO-Gen, which exposes mechanical memorization through rigorous, hierarchical probing. By decomposing policy failures, we show that a divide-and-conquer strategy combining structured data

sampling with continuous flow-matching decoders such as Pi0 outperforms discrete baselines, providing a verifiable path toward robust embodied generalization.

## Acknowledgements

National Natural Science Foundation of China (NSFC) under Grant No. 62476107.

## Impact Statement

We present LIBERO-Gen, a theory-grounded evaluation framework based on compositional generalization for assessing the reliability of Vision-Language-Action (VLA) models under controlled distribution shifts. Rather than claiming to replace real-robot evaluation, LIBERO-Gen provides a more structured and validity-oriented protocol for measuring generalization within constrained dataset settings. By systematically varying objects, scenes, instructions, and task compositions, LIBERO-Gen uncovers the Generalization Mirage: high benchmark scores can coexist with pronounced fragility under distribution shifts. Our results suggest that part of the observed performance may arise from spurious training-set correlations rather than transferable physical competence. The proposed protocols therefore serve as a diagnostic tool to better separate memorization-driven behavior from robust policy learning, while also surfacing recurring failure modes in discrete action modeling, such as Binding Collapse. Overall, our study provides empirical evidence toward narrowing the gap between simulation-based metrics and physical utility, and motivates deployment criteria that combine controlled generalization evaluation with real-world robot development.

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

# A. Theory

## A.1. Compositional Generalization

We extend the formalization of compositional generalization (CG) for Vision-Language-Action (VLA) policies by deriving sufficient conditions for its emergence and theoretical error bounds, integrating core insights from kernel theory (Lippl & Stachenfeld, 2024), general CG frameworks (Fu et al., 2024), and compositional data-generating process analysis (Wiedemer et al., 2023).

### A.1.1. SUFFICIENT CONDITIONS FOR COMPOSITIONAL GENERALIZATION

For a VLA policy $\pi_\theta$ to exhibit CG, the following conditions must hold, ensuring alignment between the policy's inductive biases, the task-context structure, and the training data's statistical properties:

**Compositional Structured Representations** The policy's latent representation $\phi(\tau, c)$ (mapping task-context tuples $x = (\tau, c_1, c_2, ..., c_n)$ to feature space) must be compositionally structured. Formally, the kernel induced by $\phi$, $K((\tau, c), (\tau', c')) = \phi(\tau, c)^T \phi(\tau', c')$, depends only on the number of identical causal factors between $(\tau, c)$ and $(\tau', c')$—where causal factors include the abstract task semantics $\tau$ and independent context factors $c_i$ (e.g., visual appearance, object identities). This ensures generalization relies on compositional similarity rather than spurious correlations. Such representations conserve compositional structure through nonlinear transforms: for random weight neural networks in the infinite-width limit, the kernel of the transformed representation depends only on the input kernel, preserving the compositional structure of the input (Lippl & Stachenfeld, 2024).

**Sufficient Support of Training Distribution** The training set $\mathcal{D}_{\text{train}}$ must have sufficient compositional support with respect to the joint task-context space $\mathcal{T} \times \mathcal{C}$, where $\mathcal{C} = \mathcal{C}_1 \times \mathcal{C}_2 \times \cdots \times \mathcal{C}_n$ denotes the joint context space. First, for every task $\tau \in \text{proj}_{\mathcal{T}}(\mathcal{D}_{\text{train}})$, there exist contexts $c^{(1)}, ..., c^{(k)} \in \mathcal{C}$ such that the subset $\{(\tau, c^{(1)}), ..., (\tau, c^{(k)})\} \subseteq \mathcal{D}_{\text{train}}$ covers all marginal contexts $c_i \in \text{proj}_{\mathcal{C}_i}(\mathcal{D}_{\text{train}})$. Second, for every context factor $c_i \in \text{proj}_{\mathcal{C}_i}(\mathcal{D}_{\text{train}})$, the training set includes pairs $(\tau^{(1)}, c)$ and $(\tau^{(2)}, c)$ (for distinct $\tau^{(1)}, \tau^{(2)}$) to disentangle task semantics from context. The support must be an open set, and for any latent configuration of a component, there exists a finite subset of training points such that the sum of the derivatives of the composition function (mapping component representations to final outputs) has full rank—ensuring sufficient variability to reconstruct component-specific behaviors without ambiguity (e.g., resolving occlusions in visual contexts by observing multiple configurations) (Wiedemer et al., 2023).

**Conjunction-Wise Additive Computation** The policy's action mapping must be conjunction-wise additive. For any test tuple $(\tau, c) \in \mathcal{D}_{\text{eval}}^{(1)}$ (composed of novel combinations of seen factors), the policy's output satisfies:

$$\pi_\theta(a \mid \tau, c) = \sum_{J \in \text{Conj}((\tau, c) \mid \mathcal{D}_{\text{train}})} f_J(\tau_J, c_J)$$

Here, $J$ denotes a subset of causal factors (task + context), $\text{Conj}((\tau, c) \mid \mathcal{D}_{\text{train}})$ is the set of factor conjunctions observed in $\mathcal{D}_{\text{train}}$ (i.e., $J \subseteq \{1, ..., n+1\}$ where there exists $(\tau^{tr}, c^{tr}) \in \mathcal{D}_{\text{train}}$ such that $\tau_J = \tau_J^{tr}$ and $c_J = c_J^{tr}$ for all indices in $J$), and $f_J$ are task-context agnostic functions mapping conjunctions of factors to action contributions. This form means the policy generalizes by combining learned factor-specific behaviors, avoiding memorization of spurious task-context pairs. For kernel models with compositionally structured representations, this additive form is a fundamental constraint—they cannot learn non-additive tasks (e.g., transitive equivalence relations requiring $A = B$ and $B = C$ implies $A = C$) as these demand reasoning beyond summing observed conjunctions (Lippl & Stachenfeld, 2024).

**Minimal Mutual Information with Spurious Rules** The policy's latent representation $\phi(\tau, c)$ must minimize mutual information with spurious correlations between input features and the target (shortcut bias) given the training distribution $\mathcal{D}_{\text{train}}$. Formally:

$$I(\phi(\tau, c); T_{\text{spurious}} \mid \mathcal{D}_{\text{train}}) \leq \delta$$

for a small $\delta > 0$, where $T_{\text{spurious}}$ denotes spurious compositional rules (e.g., associating "pick" tasks exclusively with red objects). This avoids shortcut learning, where models exploit task-irrelevant correlations that do not hold in novel combinations—such as over-reliance on context cues that predict the target in training data but fail in $\mathcal{D}_{\text{eval}}^{(1)}$ (Fu et al., 2024).

A.1.2. ERROR BOUND FOR COMPOSITIONAL GENERALIZATION

We derive a generalization error bound for $\pi_\theta$ on $\mathcal{D}_{\text{eval}}^{(1)}$, integrating IID generalization bounds and the information-theoretic analysis from (Fu et al., 2024) and (Lippl & Stachenfeld, 2024):

**Theorem A.1** Let $\pi_\theta$ be a VLA policy satisfying Conditions 1–4 above. Let $err(\pi_\theta; \mathcal{D})$ denote the expected execution error (1 - $\text{Succ}(\tau)$) of $\pi_\theta$ on distribution $\mathcal{D}$, and assume $err(\pi_\theta; \cdot)$ is $L$-bounded (i.e., $0 \leq err(\pi_\theta; \mathcal{D}) \leq L$ for all $\mathcal{D}$). Let $\text{Gen}_{IID}$ denote any IID generalization bound (e.g., Rademacher complexity, uniform stability) for $\pi_\theta$ on $\mathcal{D}_{\text{train}}$. Then:

$$\mathbb{E}_{(\tau,c)\sim\mathcal{D}_{\text{eval}}^{(1)}}\left[err(\pi_\theta; (\tau, c))\right] \leq \text{Gen}_{IID} + \kappa_n L \Phi\left(I(\phi(\tau, c); T_{\text{spurious}} \mid \mathcal{D}_{\text{train}})\right) + \text{Err}_{\text{add}} + \mathcal{O}(\epsilon_0)$$

where:

- $\kappa_n = \frac{|\mathbb{E}_{\mathcal{D}_{\text{train}}}[err(\pi_\theta;\mathcal{D}_{\text{eval}}^{(1)})-err(\pi_\theta;\mathcal{D}_{\text{train}})]|}{|\text{err}(\pi_\theta^*;\mathcal{D}_{\text{eval}}^{(1)})-\text{err}(\pi_\theta^*;\mathcal{D}_{\text{train}})|}$ quantifies data efficiency, with $\kappa_n \to 1$ as $|\mathcal{D}_{\text{train}}| \to \infty$ (capturing convergence of finite-sample training to infinite-data performance),

- $\Phi(x) = \sqrt{\min\{x/2, 1 - \exp(-x)\}}$ is an information-theoretic penalty term for shortcut bias,

- $\text{Err}_{\text{add}} = \frac{(p-2)S(1;n+1)}{1+(p-2)S(1;n+1)} \cdot \epsilon_{\text{train}}$ is the memorization leak penalty—here, $p = |\text{proj}_{\mathcal{T}}(\mathcal{D}_{\text{train}})| + \sum_{i=1}^{n} |\text{proj}_{\mathcal{C}_i}(\mathcal{D}_{\text{train}})|$ denotes the size of observed factor values, $S(1; n+1)$ is the representational salience of individual factors (normalized contribution of single-component conjunctions to the representation, with saliences $S(k; C)$ summing to 1 across all conjunction sizes $k$), and $\epsilon_{\text{train}} = \mathbb{E}_{(\tau,c)\sim\mathcal{D}_{\text{train}}}[err(\pi_\theta; (\tau, c))]$ is the training error,

- $\epsilon_0 = \max_{(\tau,c)\in\mathcal{T}\times\mathcal{C}} \min_\pi err(\pi; (\tau, c))$ is the Bayes error of the VLA task, capturing inherent task difficulty (e.g., physical constraints on action execution).

**Interpretation of Error Components** The bound consists of four key terms:

- $\text{Gen}_{IID}$ is the standard generalization error from fitting the training distribution, decreasing with larger training set size and lower model complexity.

- $\kappa_n L \Phi(I(\cdot; \cdot))$ is the penalty for shortcut bias, increasing with mutual information between the representation and spurious rules—this term vanishes as the policy avoids relying on task-irrelevant correlations.

- $\text{Err}_{\text{add}}$ is the penalty for memorization leak, where the model partially relies on full conjunctions (rather than individual components) during training, distorting generalization. This term decreases as $S(1; n + 1)$ (salience of individual factors) increases (e.g., in disentangled representations) and as training set size $p$ grows (reducing the model's need to rely on conjunctions for fitting).

- $\mathcal{O}(\epsilon_0)$ is the inherent error floor from task difficulty, representing the minimal achievable error for any VLA policy.

A.1.3. DISCUSSION

**Necessity of Conditions** Violating any condition leads to CG failure: unstructured representations enable shortcut learning by conflating spurious correlations with compositional structure; insufficient support prevents factor disentanglement (e.g., a context factor paired with only one task cannot be generalized to others); non-additive computations block generalization to novel conjunctions; and high mutual information with spurious rules degrades out-of-distribution performance.

**Practical Implications** The bound highlights that CG performance improves with three key factors: (1) larger training sets, which reduce $\text{Gen}_{IID}$ and $\kappa_n$; (2) more disentangled representations, which increase $S(1; n + 1)$ and reduce $\text{Err}_{\text{add}}$; (3) lower mutual information with spurious rules, which reduces $\Phi(I(\cdot; \cdot))$. For deep neural networks (e.g., ConvNets, ResNets, Vision Transformers), these insights hold qualitatively: deeper networks tend to encode full conjunctions (lower $S(1; C)$) and suffer more from memorization leak, while increased spatial separation between components in inputs boosts $S(1; C)$ and improves generalization. Additionally, context-dependent tasks with strong spurious correlations (e.g., context predicting most training targets) are more sensitive to architectural choices (depth, nonlinearity) that shape representational geometry, aligning with observations that shortcut bias is task-dependent.

## A.2. Domain Generalization

### A.2.1. INDEPENDENT MECHANISMS IN VLA SETTINGS

We formalize the **Independent Mechanisms (IM)** principle for VLA policies, aligning with the causal framework in (Besserve et al., 2021) and extending it to Vision-Language-Action triples. A VLA system's generative process is governed by a set of latent causal mechanisms $\{M_j\}_{j=1}^J$, where each $M_j$ corresponds to a modular component of task execution:

- $M_1$: Task semantics (e.g., goal specification, sub-task decomposition),

- $M_2$: Object dynamics (e.g., physical interactions between objects),

- $M_3$: Language grounding (e.g., mapping natural language to actionable goals),

- $M_4$: Visual appearance (e.g., object textures, lighting, camera viewpoints),

- $M_5$: Action execution (e.g., mapping policy outputs to motor commands).

The Independent Mechanisms principle imposes three key conditions on these mechanisms. First, conditional independence requires that for any two distinct mechanisms $M_j$ and $M_k$, the conditional distribution of their outputs is independent given their respective causal parents. Here, $P(\cdot)$ denotes a probability distribution, and $\mathrm{Pa}(M)$ denotes the set of causal parent nodes of mechanism $M$ in the causal graph, which means $P(M_j(\cdot) \mid \mathrm{Pa}(M_j)) \perp\!\!\!\perp P(M_k(\cdot) \mid \mathrm{Pa}(M_k))$. Second, modular invariance ensures that interventions on one mechanism such as altering visual appearance do not disrupt the functionality of other mechanisms, meaning task semantics for instance remain unchanged. Third, structural consistency dictates that all domains $e \in \mathcal{E}_{\mathrm{all}}$ share the same causal graph structure of mechanisms, with differences only arising in the instantiation of mechanism parameters. For example, parameters of the visual appearance mechanism $M_4$ may vary across lighting conditions, while parameters of the task semantics mechanism $M_1$ remain fixed for a given task $\tau$. This definition extends the work of (Besserve et al., 2021) by explicitly incorporating VLA-specific mechanisms (language grounding and action execution) and linking mechanism modularity to domain shifts. Unseen visual appearances for example correspond to intervened parameters of $M_4$.

### A.2.2. DOMAIN GENERALIZATION ERROR BOUNDS

We derive error bounds for VLA domain generalization, building on the expansion function framework by (Ye et al., 2021) and the simplicity principle by (Ge et al., 2025). Our focus is on the worst-case error across unseen domains, which captures robustness to out-of-support mechanism instantiations.

**Notation**  We denote the VLA policy as $\pi_\theta(a_t \mid o_t, l, e)$, which maps observations $o_t$, language $l$, and domain $e$ to actions $a_t$. The mechanism variation of $\pi_\theta$, denoted $\mathcal{V}(\pi_\theta, \mathcal{E}_{\mathrm{train}})$, refers to the maximum difference in policy behavior across training domains, measured by the Total Variation (TV) distance between action distributions induced by different mechanism instantiations. The expansion function $s : \mathbb{R}^+ \to \mathbb{R}^+$—as defined by (Ye et al., 2021)—characterizes how mechanism variation amplifies from training to unseen domains, with properties of being monotonically increasing, satisfying $s(x) \geq x$, and approaching 0 as $x$ approaches 0. Finally, the informativeness of the policy, denoted $\delta$, is the minimum TV distance between action distributions for distinct task semantics, ensuring the policy does not rely on spurious domain-specific cues.

**Theorem B.1 (Worst-Case Domain Generalization Error Bound)**  Suppose the VLA system satisfies Definition A.1 (Independent Mechanisms), and the domain generalization problem is $(s(\cdot), \delta)$-learnable as per (Ye et al., 2021). For a policy $\pi_\theta$ with mechanism variation $\mathcal{V}(\pi_\theta, \mathcal{E}_{\mathrm{train}}) = \varepsilon$ and informativeness $\mathcal{I}(\pi_\theta) \geq \delta$, the worst-case domain generalization error is bounded by:

$$\max_{e \in \mathcal{E}_{\mathrm{eval}}} \mathbb{E}_{(\tau, c) \sim P_e} [1 - \mathrm{Succ}(\tau)] \leq O\left( s(\varepsilon) + \frac{\log d}{\sqrt{n}} \right),$$

where $d$ represents the dimensionality of the policy's latent mechanism representations, and $n$ is the number of training samples across $\mathcal{E}_{\mathrm{train}}$. The $O(\cdot)$ term depends on $\delta$, the Lipschitz constant of the mechanism functions, and the boundedness of the action space.

**Proof Sketch**  The proof unfolds in three key steps. First, the link between mechanism variation and error is established: by virtue of IM modularity, policy error on unseen domains is determined by the deviation of unseen mechanism instantiations

from those encountered during training, a deviation captured by $s(\varepsilon)$—the expanded variation from training to evaluation domains as defined by (Ye et al., 2021). Second, the sample complexity term $\frac{\log d}{\sqrt{n}}$ arises from the statistical error associated with estimating mechanism parameters from finite training data, aligning with the simplicity principle by (Ge et al., 2025), which posits that simple mechanisms with low variation can be estimated more accurately. Third, the informativeness constraint $\mathcal{I}(\pi_\theta) \geq \delta$ ensures the policy relies on invariant task semantics rather than spurious domain cues, preventing error amplification from uninformative mechanisms as noted by (Ye et al., 2021).

**Corollary B.2 (Linear Error Bound for Linear Policies)**   When the policy's mechanism mappings are linear—such as linear language grounding or linear action projection—the expansion function takes the form $s(x) = kx$ where $k \geq 1$ is a constant, simplifying the error bound to:

$$\max_{e \in \mathcal{E}_{\text{eval}}} \mathbb{E}_{(\tau,c) \sim P_e} \left[1 - \text{Succ}(\tau)\right] \leq O\left(k\varepsilon + \frac{\log d}{\sqrt{n}}\right).$$

This result aligns with the linear top-model bound by (Ye et al., 2021), demonstrating that linear mechanisms lead to predictable error scaling with variation.

# B. Additional Experiments

## B.1. Confidence Interval Analysis for Main Results

To verify that our conclusions are not caused by evaluation randomness, we perform an additional statistical analysis based on repeated evaluations. All main results reported in Tables 1 and 2 are averaged over 50 independent evaluation runs. For each setting, we compute the sample variance and the 95% confidence interval. Unless otherwise specified, each confidence interval is reported as the mean plus/minus the half-width of the 95% confidence interval. Table 3 presents representative confidence intervals for the $\pi_0$ and $\pi_{0.5}$ results in Table 1. Across different perturbation groups, most confidence intervals are narrow, indicating that the observed performance trends are stable across repeated evaluations. This further supports the reliability of the main conclusions.

*Table 3.* Representative 95% confidence intervals for the $\pi_0$ and $\pi_{0.5}$ results in Table 1. Each entry is reported as mean $\pm$ half-width of the 95% confidence interval over 50 independent evaluation runs.

| Group | Index 0 | | Index 1 | | Index 2 | |
|---|---|---|---|---|---|---|
| | $\pi_0$ | $\pi_{0.5}$ | $\pi_0$ | $\pi_{0.5}$ | $\pi_0$ | $\pi_{0.5}$ |
| position | $0.95 \pm 0.03$ | $0.98 \pm 0.03$ | $0.07 \pm 0.02$ | $0.62 \pm 0.04$ | $0.05 \pm 0.02$ | $0.25 \pm 0.05$ |
| task | $0.98 \pm 0.02$ | $0.99 \pm 0.01$ | $0.14 \pm 0.03$ | $0.20 \pm 0.03$ | $0.40 \pm 0.05$ | $0.52 \pm 0.06$ |
| object | $0.91 \pm 0.03$ | $0.99 \pm 0.02$ | $0.98 \pm 0.02$ | $1.00 \pm 0.00$ | $0.88 \pm 0.04$ | $1.00 \pm 0.01$ |
| texture | $0.97 \pm 0.02$ | $1.00 \pm 0.01$ | $0.96 \pm 0.02$ | $1.00 \pm 0.00$ | $0.99 \pm 0.02$ | $1.00 \pm 0.00$ |
| language | $0.99 \pm 0.02$ | $1.00 \pm 0.01$ | $0.98 \pm 0.02$ | $1.00 \pm 0.01$ | $0.95 \pm 0.04$ | $0.99 \pm 0.01$ |

## B.2. Statistical Significance of Stair vs. Random

We further evaluate whether the advantage of STAIR over RANDOM in Fig. 8 is statistically reliable. Tables 4 and 5 summarize the mean, sample variance, and 95% confidence interval under the position and task settings, respectively.

*Table 4.* Statistical comparison between STAIR and RANDOM under the position setting.

| Method | Mean | Sample Variance | 95% Confidence Interval |
|---|---|---|---|
| $\pi_{0.5}$-STAIR | 0.64 | $5.000 \times 10^{-3}$ | $[0.607, 0.673]$ |
| $\pi_{0.5}$-RANDOM | 0.52 | $2.000 \times 10^{-3}$ | $[0.499, 0.541]$ |
| OpenVLA-STAIR | 0.26 | $1.286 \times 10^{-4}$ | $[0.2547, 0.2653]$ |
| OpenVLA-RANDOM | 0.02 | $4.000 \times 10^{-7}$ | $[0.0197, 0.0203]$ |

The results show that STAIR consistently outperforms RANDOM. For $\pi_{0.5}$, the improvement is statistically significant under both the position setting and the task setting, with $p = 3.24 \times 10^{-7}$ and $p = 1.91 \times 10^{-4}$, respectively. For OpenVLA, the improvement is also statistically significant under the position setting, with $p = 5.37 \times 10^{-27}$. Under the task setting, both

*Table 5.* Statistical comparison between STAIR and RANDOM under the task setting.

| Method | Mean | Sample Variance | 95% Confidence Interval |
|---|---|---|---|
| $\pi_{0.5}$-STAIR | 0.21 | $1.000 \times 10^{-3}$ | $[0.195, 0.225]$ |
| $\pi_{0.5}$-RANDOM | 0.15 | $3.000 \times 10^{-3}$ | $[0.124, 0.176]$ |
| OpenVLA-STAIR | 0.00 | 0.000 | $[0.000, 0.000]$ |
| OpenVLA-RANDOM | 0.00 | 0.000 | $[0.000, 0.000]$ |

OpenVLA variants achieve zero success rate across all 50 runs, so we do not report a significance test for this comparison. Overall, these results confirm that the advantage of STAIR is statistically reliable rather than an artifact of random evaluation noise.

## C. LIBERO-Gen Design Details

This appendix delineates the granular specifications and data generation protocols for the five evaluation suites constituting the LIBERO-Gen benchmark. We detail the precise construction of the training distributions ($\mathcal{D}_{train}$) and the hierarchical evaluation regimes ($k \in \{0, 1, 2\}$) to ensure the reproducibility of the "Stair" and "L-shape" sampling strategies discussed in Section 3.

### C.1. LIBEROGEN-Texture

In this suite, we evaluate the visual invariance of VLA policies by systematically varying the background textures (specifically the floor appearance) while keeping task semantics and object layouts constant. We utilize the 10 atomic object manipulation tasks from LIBERO-Object as the foundation.

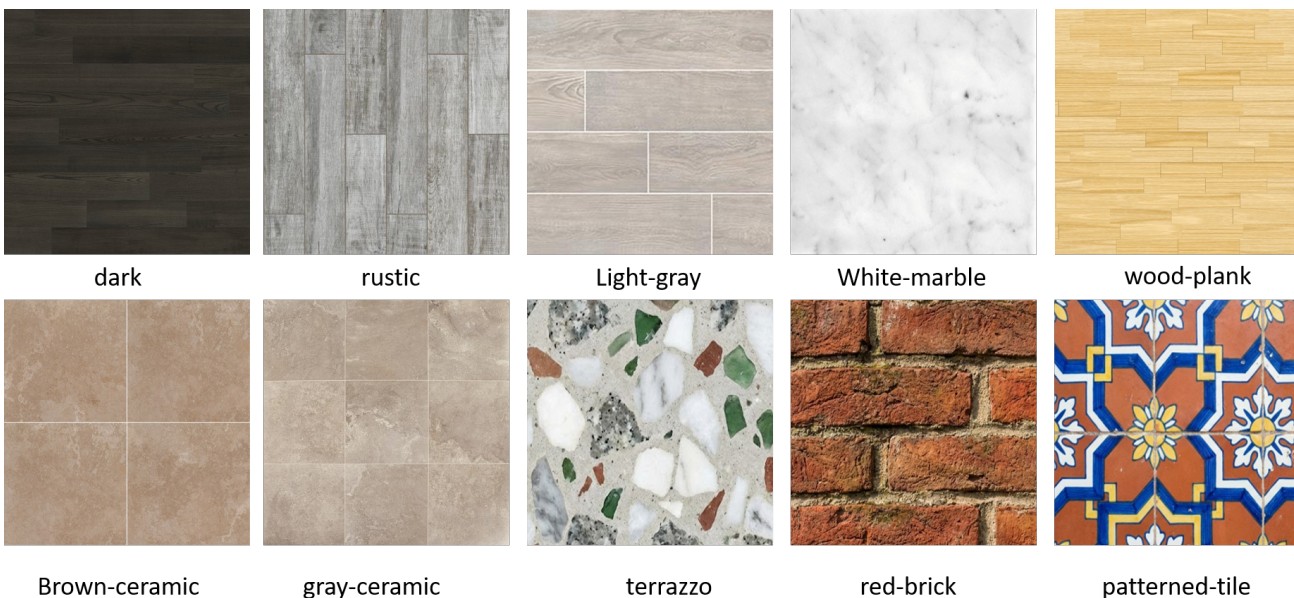

| dark | rustic | Light-gray | White-marble | wood-plank |

| Brown-ceramic | gray-ceramic | terrazzo | red-brick | patterned-tile |

*Figure 9.* Overview of Pattern Designs.

**Texture Assets.** We curated two disjoint sets of floor texture assets to strictly separate in-distribution (ID) visual features from out-of-distribution (OOD) features, as shown in Figure 9.

- **Seen Textures ($\mathcal{C}_{seen}$).** Used for training and compositional generalization. The set includes five distinct textures: (1) dark (Dark wood flooring), (2) rustic (Rustic wooden planks), (3) light-gray (Smooth light gray tiling), (4) white-marble (Polished white marble), and (5) wood-plank (Standard wooden deck).

- **Unseen Textures ($\mathcal{C}_{unseen}$).** Used exclusively for domain generalization ($k = 2$). The set includes: (1) brown-ceramic, (2) gray-ceramic, (3) patterned-tile, (4) terrazzo, and (5) red-brick.

**Training Distribution): "Stair" Sampling Protocol.** To induce spurious correlations while ensuring sufficient support (as defined in Appendix A), we employ a "Stair" sampling strategy. We partition the 10 tasks into 5 groups ($\mathcal{G}_1 \ldots \mathcal{G}_5$) and bind each group to a sliding window of textures. Specifically, each task group is trained on two adjacent textures from the ordered set $\mathcal{C}_{seen}$, forming a diagonal band in the task-texture matrix, as shown in Figure 10. The exact training configurations are detailed in Table 6.

*Table 6.* Training Distribution ($\mathcal{D}_{train}$) for LIBEROGEN-Texture with Stair Sampling. Each task group is associated with a sliding window of two textures.

| Task Group | Tasks Included (Full Instructions) | Associated Textures |
|---|---|---|
| **Group 1** | 1. pick_up_the_alphabet_soup_and_place_it_in_the_basket
2. pick_up_the_bbq_sauce_and_place_it_in_the_basket | `dark`, `rustic` |
| **Group 2** | 3. pick_up_the_butter_and_place_it_in_the_basket
4. pick_up_the_chocolate_pudding_and_place_it_in_the_basket | `rustic`, `light-gray` |
| **Group 3** | 5. pick_up_the_cream_cheese_and_place_it_in_the_basket
6. pick_up_the_ketchup_and_place_it_in_the_basket | `light-gray`, `white-marble` |
| **Group 4** | 7. pick_up_the_milk_and_place_it_in_the_basket
8. pick_up_the_orange_juice_and_place_it_in_the_basket | `white-marble`, `wood-plank` |
| **Group 5** | 9. pick_up_the_salad_dressing_and_place_it_in_the_basket
10. pick_up_the_tomato_sauce_and_place_it_in_the_basket | `wood-plank`, `dark` |

**Evaluation Distribution.** It is important to note that while a robust evaluation in a controlled setting could theoretically involve randomized combinations of tasks and textures, such randomness can introduce variance that hinders direct model comparison. To ensure strict experimental reproducibility and provide a standardized reference for comparison in this paper, we explicitly list the specific evaluation set used in our experiments in Table 7.

*Table 7.* Fixed Evaluation Set for LIBEROGEN-Texture. We report the specific background textures used for Compositional Generalization (CG, $k = 1$) and Domain Generalization (DG, $k = 2$) for each task to guarantee reproducibility.

| Task Instruction | CG Texture ($k = 1$) | DG Texture ($k = 2$) |
|---|---|---|
| pick_up_the_alphabet_soup_and_place_it_in_the_basket | `rustic` | `brown-ceramic` |
| pick_up_the_cream_cheese_and_place_it_in_the_basket | `white-marble` | `brown-ceramic` |
| pick_up_the_salad_dressing_and_place_it_in_the_basket | `dark` | `tile_grigia_caldera` |
| pick_up_the_bbq_sauce_and_place_it_in_the_basket | `light-gray` | `gray-ceramic` |
| pick_up_the_ketchup_and_place_it_in_the_basket | `wood-plank` | `gray-ceramic` |
| pick_up_the_tomato_sauce_and_place_it_in_the_basket | `rustic` | `gray-ceramic` |
| pick_up_the_butter_and_place_it_in_the_basket | `rustic` | `tile_grigia_caldera` |
| pick_up_the_milk_and_place_it_in_the_basket | `wood-plank` | `gray-ceramic` |
| pick_up_the_chocolate_pudding_and_place_it_in_the_basket | `white-marble` | `tile_grigia_caldera` |
| pick_up_the_orange_juice_and_place_it_in_the_basket | `rustic` | `gray-ceramic` |

### C.2. LIBEROGEN-Object

In this suite, we evaluate the robustness of VLA policies against visual distractors (task-irrelevant objects) while keeping task semantics and object layouts constant. Similar to the texture suite, we utilize the 10 atomic object manipulation tasks from LIBERO-Object as the foundation.

**Distractor Assets.** We curated two disjoint sets of object assets to strictly separate in-distribution (ID) visual features from out-of-distribution (OOD) features.

- **Seen Distractors ($\mathcal{C}_{seen}$).** Used for training and compositional generalization. We define 5 distinct distractor configurations, where each configuration consists of 5 instances of a specific object category: (1) akita_black_bowl, (2) glazed_rim_porcelain_ramekin, (3) moka_pot, (4) porcelain_mug, and (5) wine_bottle.

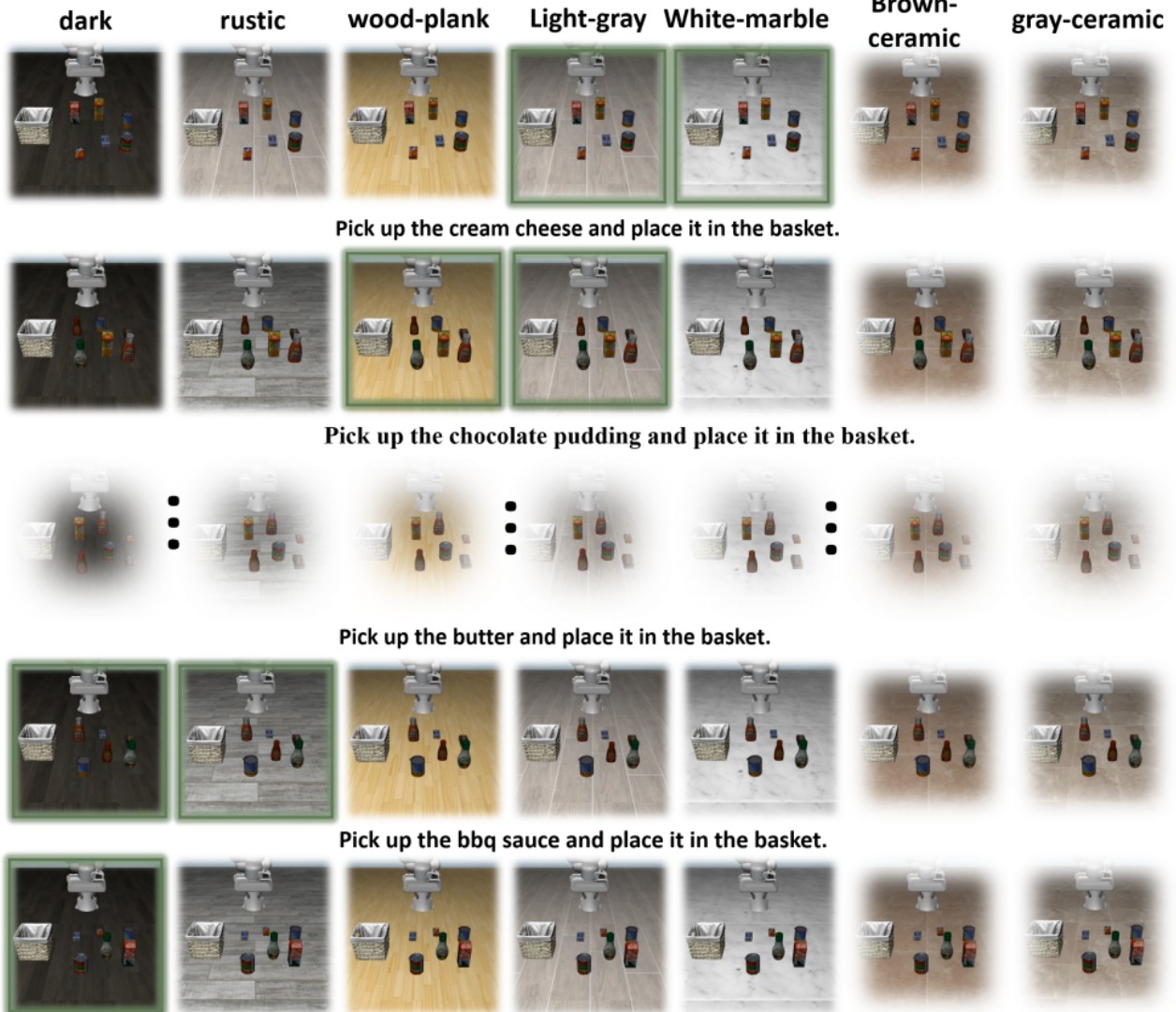

*Figure 10.* LIBEROGEN-Texture.

- **Unseen Distractors** ($\mathcal{C}_{unseen}$). Used exclusively for domain generalization ($k = 2$). The pool includes 10 object categories: alphabet_soup, bbq_sauce, butter, chocolate_pudding, cream_cheese, ketchup, milk, orange_juice, salad_dressing, and tomato_sauce.

**Training Distribution: "Stair" Sampling Protocol.** To induce spurious correlations while ensuring sufficient support, we employ the same "Stair" sampling strategy used in the texture suite. We partition the 10 tasks into 5 groups ($\mathcal{G}_1 \ldots \mathcal{G}_5$) and bind each group to a sliding window of distractor configurations. Specifically, each task group is trained with two adjacent distractor configurations from the ordered set $\mathcal{C}_{seen}$. The exact training configurations are detailed in Table 8.

**Evaluation Distribution.** Consistent with our protocol for visual textures, we establish a fixed evaluation set to ensure reproducibility. For Compositional Generalization ($k = 1$), tasks are paired with seen distractor configurations that were not present in their training window. For Domain Generalization ($k = 2$), each task is evaluated against a unique, fixed combination of 5 objects randomly sampled from the $\mathcal{C}_{unseen}$ pool. These fixed sets are listed in Table 9.

*Table 8.* Training Distribution ($\mathcal{D}_{train}$) for LIBEROGEN-Object with Stair Sampling. Each task group is associated with a sliding window of two distractor configurations (each containing 5 instances of the specified object).

| Task Group | Tasks Included (Full Instructions) | Associated Distractors (5x) |
|---|---|---|
| **Group 1** | 1. pick_up_the_alphabet_soup_and_place_it_in_the_basket 
 2. pick_up_the_bbq_sauce_and_place_it_in_the_basket | `akita_black_bowl`, `glazed_rim_porcelain_ramekin` |
| **Group 2** | 3. pick_up_the_butter_and_place_it_in_the_basket 
 4. pick_up_the_chocolate_pudding_and_place_it_in_the_basket | `glazed_rim_porcelain_ramekin`, `moka_pot` |
| **Group 3** | 5. pick_up_the_cream_cheese_and_place_it_in_the_basket 
 6. pick_up_the_ketchup_and_place_it_in_the_basket | `moka_pot`, `porcelain_mug` |
| **Group 4** | 7. pick_up_the_milk_and_place_it_in_the_basket 
 8. pick_up_the_orange_juice_and_place_it_in_the_basket.bddl | `porcelain_mug`, `wine_bottle` |
| **Group 5** | 9. pick_up_the_salad_dressing_and_place_it_in_the_basket.bddl 
 10. pick_up_the_tomato_sauce_and_place_it_in_the_basket | `wine_bottle`, `akita_black_bowl` |

### C.3. LIBEROGEN-Language

In this suite, we evaluate the robustness of VLA policies to linguistic variations, specifically testing whether agents rely on syntactic pattern matching or achieve true semantic grounding. We utilize the same 10 atomic tasks from LIBERO-Object but systematically vary the instruction phrasing.

**Language Assets.** We constructed two disjoint sets of instructions to separate in-distribution (ID) syntactic patterns from out-of-distribution (OOD) semantic and noisy variations.

- **Seen Templates ($\mathcal{C}_{seen}$).** Used for training and compositional generalization. We designed 5 semantically equivalent but syntactically distinct templates, where $\{x\}$ denotes the object and $\{y\}$ denotes the receptacle (e.g., "the basket"): (1) "Pick $\{x\}$ and put it in $\{y\}$." (2) "Pick up $\{x\}$, then place it in $\{y\}$." (3) "Move $\{x\}$ into $\{y\}$." (4) "Put $\{x\}$ into $\{y\}$." (5) "$\{x\}$ should be picked up and placed in $\{y\}$."

- **Unseen Variations ($\mathcal{C}_{unseen}$).** Used exclusively for domain generalization ($k = 2$). For each task, we generated 10 novel instructions using an LLM (GPT-4), comprising 5 *Semantic Paraphrases* (diverse vocabulary/structure) and 5 *Noisy/Irrelevant Contexts* (embedded in conversational filler or environmental descriptions).

**Training Distribution: "Stair" Sampling Protocol.** To prevent the model from observing all syntactic variations for every task, we employ a "Stair" sampling strategy. We partition the 10 tasks into 5 groups ($\mathcal{G}_1 \ldots \mathcal{G}_5$) and bind each group to a sliding window of instruction templates. Specifically, each task group is trained on only two specific adjacent templates from $\mathcal{C}_{seen}$. The exact training configurations are detailed in Table 10.

**Evaluation Distribution.** To ensure reproducibility, we define a fixed evaluation set. For Compositional Generalization ($k = 1$), we evaluate tasks using a seen template that was *not* in their training window (e.g., Group 1 is evaluated on Template 3). For Domain Generalization ($k = 2$), we evaluate on the fixed set of 10 unseen variations per task. Table 11 lists the specific CG template and representative DG examples used.

### C.4. LIBEROGEN-Position

In this suite, we evaluate the spatial reasoning capabilities of VLA policies, specifically testing whether agents can extrapolate learned spatial relationships to novel configurations. Unlike previous suites that vary semantic factors, here we fix the task semantic to "Pick up the bowl and place it on the plate" and treat the spatial locations of the *Bowl* (Start Position) and *Plate* (Target Position) as the independent context variables.

**Spatial Assets.** We define the workspace using a set of semantic regions for in-distribution (grid-based) tasks and continuous coordinates for out-of-distribution tasks.

- **Seen Discrete Regions ($\mathcal{C}_{seen}$).** We identified 9 distinct semantic regions on the tabletop to construct a discretized $9 \times 9$ grid. Let $\mathcal{L} = \{L_0, \ldots, L_8\}$ denote these locations: (1) $L_0$: `main_table_table_center` (Canonical Anchor), (2) $L_1$: `wooden_cabinet_1_top_region`, (3) $L_2$: `main_table_next_to_box_region`, (4) $L_3$: `main_table_next_to_plate_region`, (5) $L_4$: `main_table_next_to_ramekin_region`, (6) $L_5$: `cookies_1` (On top of cookies), (7) $L_6$: `glazed_rim_porcelain_ramekin_1`, (8) $L_7$:

*Table 9.* Fixed Evaluation Set for LIBEROGEN-Object. We report the specific distractor configurations used for Compositional Generalization (CG, $k = 1$) and Domain Generalization (DG, $k = 2$). For DG, each entry represents a set of 5 distractor objects present in the scene.

| Task Instruction | CG Distractors ($k = 1$) | DG Distractors ($k = 2$) |
|---|---|---|
| pick_up_the_alphabet_soup_and_place_it_in_the_basket | glazed_rim_porcelain_ramekin (5x) | milk, butter, ketchup, bbq_sauce, cream_cheese |
| pick_up_the_cream_cheese_and_place_it_in_the_basket | akita_black_bowl (5x) | orange_juice, tomato_sauce, salad_dressing, chocolate_pudding, alphabet_soup |
| pick_up_the_salad_dressing_and_place_it_in_the_basket | porcelain_mug (5x) | butter, cream_cheese, milk, tomato_sauce, bbq_sauce |
| pick_up_the_bbq_sauce_and_place_it_in_the_basket | wine_bottle (5x) | ketchup, orange_juice, salad_dressing, alphabet_soup, chocolate_pudding |
| pick_up_the_ketchup_and_place_it_in_the_basket | wine_bottle (5x) | milk, bbq_sauce, butter, cream_cheese, tomato_sauce |
| pick_up_the_tomato_sauce_and_place_it_in_the_basket | porcelain_mug (5x) | orange_juice, ketchup, alphabet_soup, salad_dressing, milk |
| pick_up_the_butter_and_place_it_in_the_basket | glazed_rim_porcelain_ramekin (5x) | chocolate_pudding, butter, cream_cheese, bbq_sauce, orange_juice |
| pick_up_the_milk_and_place_it_in_the_basket | moka_pot (5x) | tomato_sauce, salad_dressing, ketchup, alphabet_soup, milk |
| pick_up_the_chocolate_pudding_and_place_it_in_the_basket | moka_pot (5x) | bbq_sauce, cream_cheese, orange_juice, butter, chocolate_pudding |
| pick_up_the_orange_juice_and_place_it_in_the_basket | glazed_rim_porcelain_ramekin (5x) | salad_dressing, tomato_sauce, milk, ketchup, alphabet_soup |

*Table 10.* Training Distribution ($\mathcal{D}_{train}$) for LIBEROGEN-Language with Stair Sampling. Each task group is strictly bound to a sliding window of two syntactic templates.

| Task Group | Tasks Included | Associated Templates |
|---|---|---|
| **Group 1** | 1. alphabet_soup
2. cream_cheese | 1. Pick $\{x\}$ and put it in $\{y\}$.
2. Pick up $\{x\}$, then place it in $\{y\}$. |
| **Group 2** | 3. salad_dressing
4. bbq_sauce | 2. Pick up $\{x\}$, then place it in $\{y\}$.
3. Move $\{x\}$ into $\{y\}$. |
| **Group 3** | 5. ketchup
6. tomato_sauce | 3. Move $\{x\}$ into $\{y\}$.
4. Put $\{x\}$ into $\{y\}$. |
| **Group 4** | 7. butter
8. milk | 4. Put $\{x\}$ into $\{y\}$.
5. $\{x\}$ should be picked up... |
| **Group 5** | 9. chocolate_pudding
10. orange_juice | 5. $\{x\}$ should be picked up...
1. Pick $\{x\}$ and put it in $\{y\}$. |

flat_stove_1_cook_region, and (9) $L_8$: wooden_cabinet_1_top_side.

- **Unseen Continuous Regions** ($\mathcal{C}_{unseen}$). Used exclusively for domain generalization ($k = 2$). Instead of adhering to the fixed semantic regions, object positions are sampled uniformly from the continuous tabletop surface $(x, y) \in \mathbb{R}^2$, excluding the specific coordinates of the grid centers defined in $\mathcal{C}_{seen}$.

**Training Distribution: "L-Shaped" Sampling Protocol.**
To test compositional generalization, we employ an "L-Shaped" sampling strategy that creates a sparse training support. We utilize $L_0$ (main_table_table_center) as the pivot location. The training set $\mathcal{D}_{train}$ consists of

20

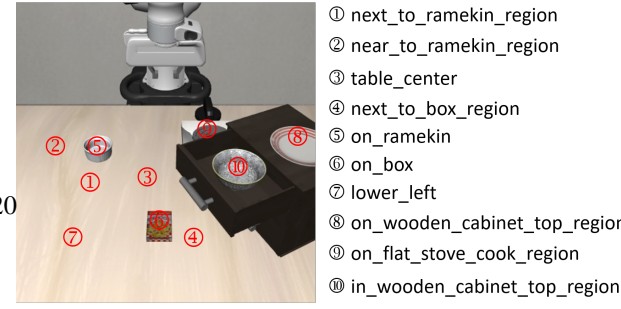

① next_to_ramekin_region
② near_to_ramekin_region
③ table_center
④ next_to_box_region
⑤ on_ramekin
⑥ on_box
⑦ lower_left
⑧ on_wooden_cabinet_top_region
⑨ on_flat_stove_cook_region
⑩ in_wooden_cabinet_top_region

*Figure 13.* Example of Position.

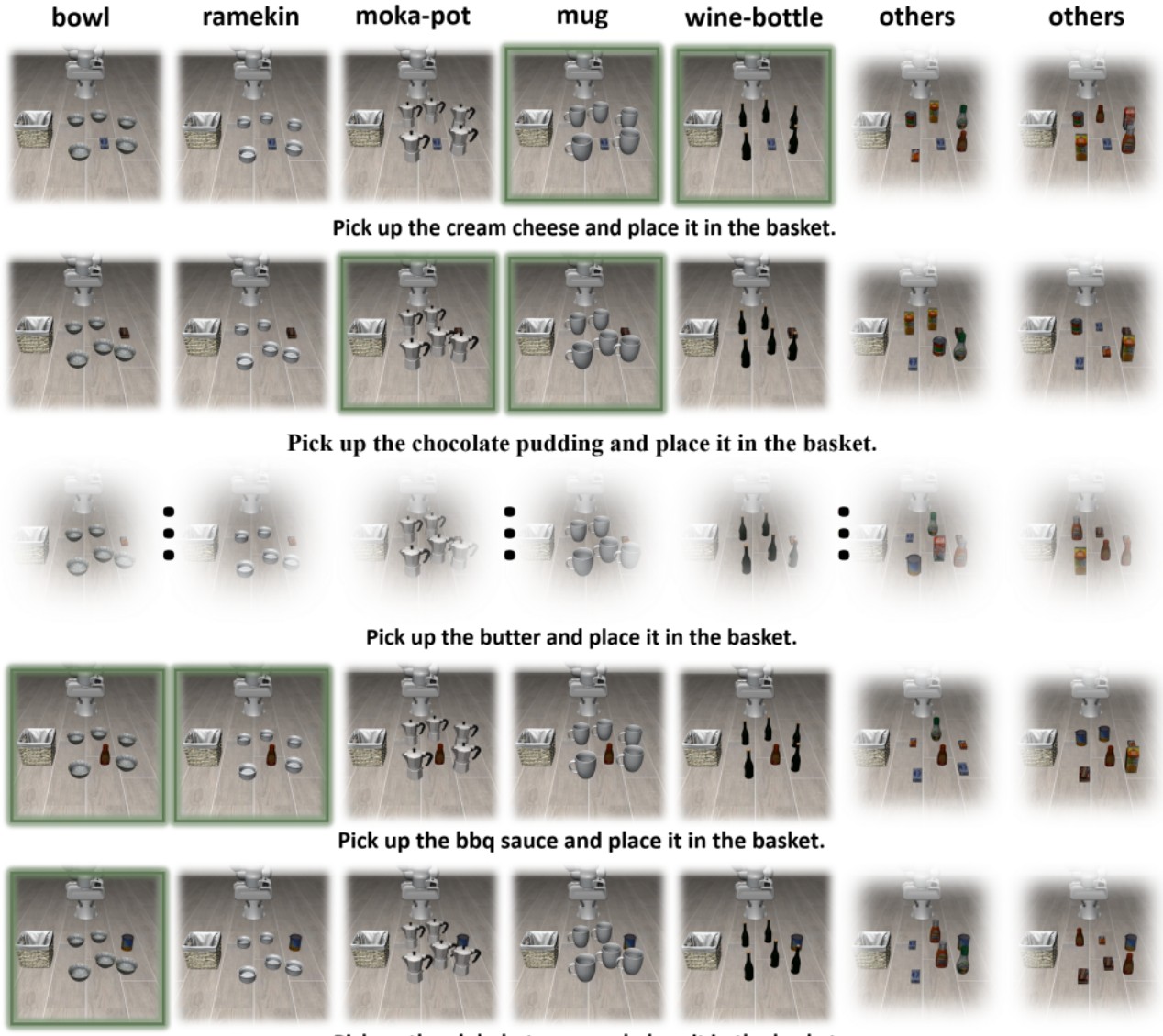

*Figure 11.* LIBEROGEN-Object.

the union of two subsets: (1) **Row Support**: The Plate is fixed at $L_0$, while the Bowl varies across all discrete locations $\{L_0 \ldots L_8\}$. (2) **Column Support**: The Bowl is fixed at $L_0$, while the Plate varies across the remaining locations $\{L_1 \ldots L_8\}$. This structure ensures the model observes every location individually but only in restricted combinations.

**Evaluation Distribution.** We establish a fixed evaluation set to ensure reproducibility.

- **CG** ($k = 1$): We evaluate on "Off-Axis" grid combinations where both the Bowl and Plate are at non-center locations (e.g., Bowl at $L_1$, Plate at $L_2$). These pairs require the model to recombine spatial knowledge learned separately during training.

- **DG** ($k = 2$): We evaluate on 10 fixed random seeds where Bowl and Plate positions are sampled continuously from the table surface, ensuring no collision with training grid points.

*Table 11.* Fixed Evaluation Set for LIBEROGEN-Language. For DG ($k = 2$), we evaluate on a fixed set of 10 variations per task; we show one representative *Semantic* and one *Noisy* example below.

| Target Object | CG Template ($k = 1$) | DG Variations ($k = 2$) [Examples] |
|---|---|---|
| alphabet_soup | Move {x} into {y}. | **S:** Transfer the can of alphabet soup into the basket. 
 **N:** It looks cloudy outside... Just grab the alphabet soup... |
| cream_cheese | Move {x} into {y}. | **S:** Transport the cream cheese to the basket. 
 **N:** They didn't have the other brand... Put the cream cheese... |
| salad_dressing | Put {x} into {y}. | **S:** Place the bottle of salad dressing into the basket. 
 **N:** I really don't feel like eating greens, but put the... |
| bbq_sauce | Put {x} into {y}. | **S:** Locate the bbq sauce and set it down inside the basket. 
 **N:** I completely forgot to clean the grill... take the bbq sauce... |
| ketchup | {x} should be... | **S:** Don't forget the condiment; place the ketchup in the basket. 
 **N:** Fries are just not the same without it. Place the ketchup... |
| tomato_sauce | {x} should be... | **S:** Take the can of tomato sauce and move it to the basket. 
 **N:** Be careful, glass breaks easily. Put the tomato sauce... |
| butter | Pick {x} and put... | **S:** Shift the butter into the basket. 
 **N:** I'm trying to cut down on calories... Put the butter... |
| milk | Pick {x} and put... | **S:** Transfer the milk to the basket. 
 **N:** My coffee was terrible... Please put the milk in the basket. |
| chocolate_pudding | Pick up {x}, then... | **S:** Retrieve the chocolate pudding and add it to the items... 
 **N:** It's not for me... Take the chocolate pudding and insert... |
| orange_juice | Pick up {x}, then... | **S:** Deposit the orange juice into the basket. 
 **N:** The doctor said I need more Vitamin C. Pick the orange... |

## C.5. LIBEROGEN-Task

In this suite, we evaluate the combinatorial generalization of VLA policies, specifically testing whether agents can compose learned atomic skills into novel long-horizon sequences or generalize to entirely new task semantics.

**Task Assets.** We define the task space based on the 10 manipulatable objects from LIBERO-Object. Let $\mathcal{O}$ be the set of objects in LIBERO-Object, and $\mathcal{P} = \mathcal{O} \times \mathcal{O}$ denote all possible object pairs.

- **Atomic Tasks** ($\mathcal{T}_{atomic}$). Corresponding to LIBERO-Object tasks, with semantics $\tau = (o)$ for each $o \in \mathcal{O}$. These are the foundational "Pick-and-Place" primitives: "Pick up the [Object] and place it in the basket". There are 10 such tasks corresponding to the 10 objects in $\mathcal{O}$.

- **Seen Composites** ($\mathcal{T}_{seen\_comp}$). Corresponding to LIBERO-10 tasks, with semantics $\tau = (o_i, o_j) \in \mathcal{P}_{seen} \subset \mathcal{P}$, where $\mathcal{P}_{seen}$ denotes the set of observed object pairs during training. We select 3 specific sequential pairs (i.e., $\mathcal{P}_{seen}$ contains 3 pairs) to serve as the sparse support for compositionality (e.g., "Pick $o_i$ then Pick $o_j$").

- **Unseen Semantics** ($\mathcal{T}_{ood}$). Used for Domain Generalization ($k = 2$). These are paired-object tasks absent from any training composite task, requiring extrapolation beyond observed compositions and task structures, rather than introducing new predicates.

**Training Distribution: Sparse Compositional Support.** Unlike the "Stair" sampling used in visual suites, the task dimension employs a "Sparse Union" strategy. The training set $\mathcal{D}_{train}$ comprises all 10 Atomic Tasks plus exactly 3 specific Composite Tasks. This setup tests if the model can infer the general rule of composition from minimal examples. The exact training set is listed in Table 12.

**Evaluation Distribution.** We establish a fixed evaluation set to ensure reproducibility.

- **CG** ($k = 1$): We evaluate the policy on novel combinations of the seen atomic tasks. These are pairs $(o_i, o_j) \notin \mathcal{P}_{seen}$ formed by recombining objects from $\mathcal{O}$ into arbitrary pairs that were never sequenced together during training (e.g.,

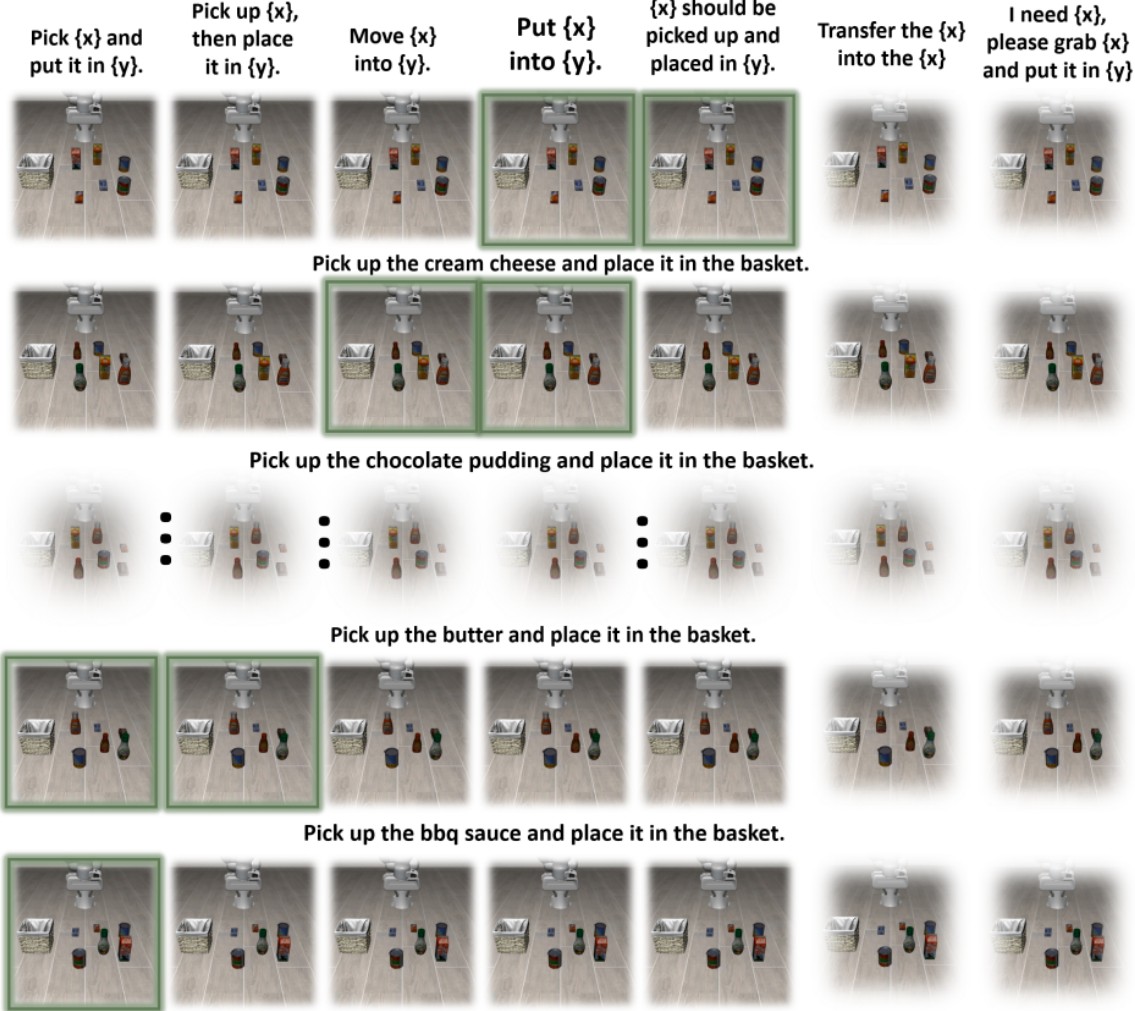

*Figure 12.* LIBEROGEN-Language.

"Pick Ketchup then Pick Soup").

- **DG** ($k = 2$): We test the policy on paired-object tasks absent from any training composite task, requiring extrapolation beyond observed compositions and task structures—going beyond mere recombination of seen atomic skills.

## D. Fault Examples

We visualize representative generalization failure cases in Figure 15.

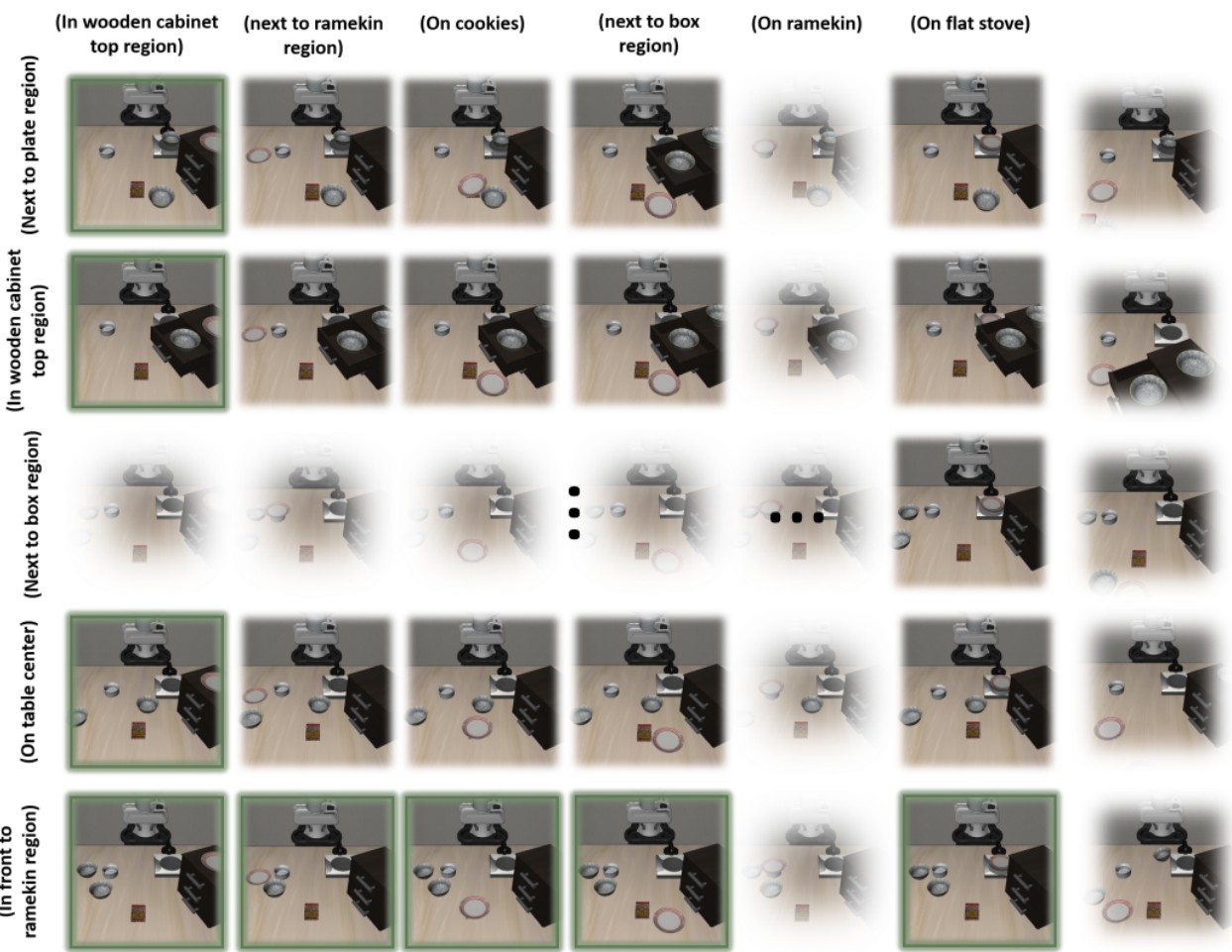

*Figure 14.* LIBEROGEN-Position.

*Table 12.* Training Distribution ($\mathcal{D}_{train}$) for LIBEROGEN-Task. Includes all atomic tasks and a sparse set of 3 composite chains.

| Task Type | Included Tasks (Full Instructions) |
| --- | --- |
| **Atomic (10x)** | 1. Pick up the alphabet soup and place it in the basket.
2. Pick up the cream cheese and place it in the basket.
...(Includes all 10 objects defined in LIBERO-Object)...
10. Pick up the orange juice and place it in the basket. |
| **Seen Composites (3x)** | 11. Pick up the alphabet soup and place it in the basket, then pick up the cream cheese and place it in the basket.
12. Pick up the salad dressing and place it in the basket, then pick up the bbq sauce and place it in the basket.
13. Pick up the butter and place it in the basket, then pick up the milk and place it in the basket. |

*Table 13.* Fixed Evaluation Set for LIBEROGEN-Task. CG ($k = 1$) tests novel sequential compositions. DG ($k = 2$) tests the unseen object-pairs.

| ID | CG Sequences ($k = 1$) [Novel Pairs] | DG Semantics ($k = 2$) [Unseen Objects] |
|---|---|---|
| 1 | Pick orange_juice then Pick alphabet_soup | Pick wine_bottle then Pick alphabet_soup |
| 2 | Pick chocolate_pudding then Pick milk | Pick alphabet_soup then Pick mug |
| 3 | Pick alphabet_soup then Pick tomato_sauce | Pick black_bowl then Pick butter |
| 4 | Pick cream_cheese then Pick alphabet_soup | Pick black_bowl then Pick tomato_sauce |
| 5 | Pick ketchup then Pick tomato_sauce | Pick both_cream_cheese |
| 6 | Pick milk then Pick butter | Pick moka_pot then Pick butter |
| 7 | Pick cream_cheese then Pick milk | Pick tomato_sauce then Pick moka_pot |
| 8 | Pick milk then Pick tomato_sauce | Pick plate then Pick cream_cheese |
| 9 | Pick orange_juice then Pick butter | Pick mug then Pick cream_cheese |
| 10 | Pick tomato_sauce then Pick ketchup | Pick wine_bottle then Pick tomato_sauce |

Distractor – changes to unrelated objects results in the policy being unable to execute actions with precision

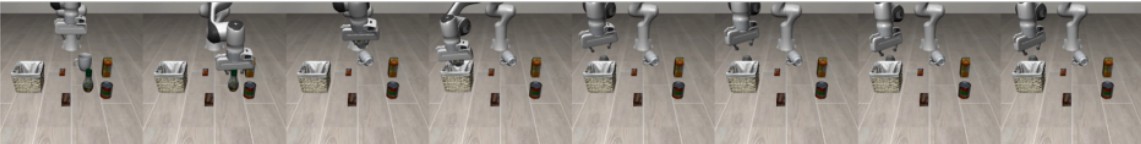

Position – altered location of target object leads to policy manipulating a wrong target

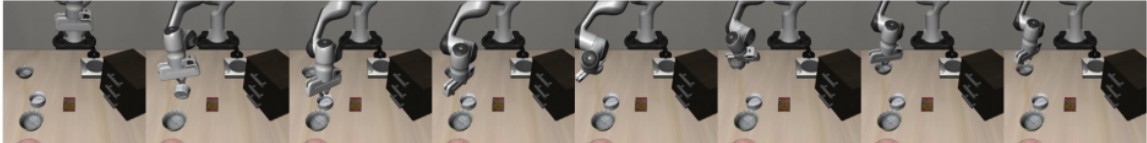

Task – policy fails to formulate an action plan when confronted with a new goal within the original environment

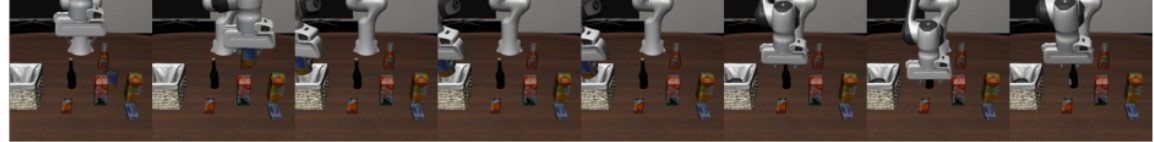

*Figure 15.* Examples of Pi0 and Pi0.5.

