# OpenReview forum: "Dismantling the Illusion of Vision-Language-Action Models Competence via Explicit Distributional Shifts"
_ICML.cc/2026/Conference — ICML 2026 regular_

### Official Review · Reviewer_3f33 · 2026-03-02

**Soundness:** 3
**Presentation:** 3
**Significance:** 3
**Originality:** 3
**Overall Recommendation:** 4
**Confidence:** 4

**Summary:**

LIBERO-Gen is a diagnostic benchmark for vision-language-action (VLA) models that evaluates compositionality and domain generalization through controllable, factorized distribution shifts along five dimensions, including background, language, distractors, task semantics, and spatial configuration. The authors formalize compositionality and domain generalization for VLA policies. Empirically, they show that several VLA models that score highly on LIBERO can break down under moderate semantic or spatial shifts, revealing failure modes such as perceptual instability and action binding collapse. Overall, LIBERO-Gen differentiates model capabilities that standard metrics may conflate, and provides evidence that structured sampling and continuous-action modeling (e.g., Pi0.5) can partially mitigate such brittleness.

**Compliance With Llm Reviewing Policy:**

Affirmed.

**Key Questions For Authors:**

(1) DG in the current benchmark primarily focuses on unseen textures and appearance-domain shifts. Do the authors plan to extend DG to broader real-world domain variations such as camera viewpoint changes, lighting conditions, sensor noise, camera intrinsics, or changes in object/device appearance? If so, do the authors have any preliminary results or insights to share?

(2) Given the paper’s strong motivation around deployment relevance, could the authors provide even a small-scale sim-to-real validation to test whether LIBERO-Gen scores correlate with hardware performance better than LIBERO?

(3) Could the authors report results for the ``Random vs.\ Stair'' comparison across multiple random seeds, including mean and variance or confidence intervals, and provide statistical significance tests to substantiate the robustness of the observed gains?

**Limitations:**

Yes

**Strengths And Weaknesses:**

Strengths:
(1) The benchmark redefines evaluation under an explicit factorized distribution-shift setting, offering a clear decomposition of controllable factors (background, language, distractors, task semantics, and spatial configuration) and a hierarchical evaluation protocol (ID, CG, and DG) to systematically assess compositional and domain generalization in VLA models.

(2) The results reveal clear capability stratification across models. In particular, some models perform strongly on language-oriented tasks, yet remain substantially more brittle when robust spatial generalization and physical grounding are required, exhibiting a notable separation between language competence and spatial robustness.

(3) It provides actionable insights for method development, highlighting the benefits of continuous action modeling and structured data topology, and may inform future VLA training and evaluation protocols.

Weaknesses:
(1) Domain generalization (DG) in this paper is primarily operationalized as robustness to unseen appearance domains outside the training support (e.g., unseen textures and materials). However, because this setting mainly captures appearance-level domain shifts, broader real-world domain variations such as viewpoint changes, complex lighting conditions, and sensor noise are not systematically incorporated into the evaluation, which may underestimate the difficulty of DG in practical deployments.

(2) Although the paper repeatedly motivates its study by aiming to close the gap between simulation metrics and real-world deployment utility and to support reliable deployment, the current experiments and conclusions are primarily grounded in simulation benchmarks and still lack external validation that aligns with real robot hardware or more realistic sensor and physical perturbations; consequently, the deployment relevance remains less convincing.

(3) The paper presents a controlled comparison between random sampling and structured Stair sampling under the same data budget, and reports that Stair yields noticeable gains. However, this conclusion is currently supported by a relatively limited set of comparisons and lacks more systematic statistical evidence, such as results across multiple random seeds with variance or confidence intervals, significance testing, and stronger or more diverse sampling baselines beyond random.

---

> ### Author Rebuttal · Authors · 2026-03-31
>
> We sincerely thank the reviewer for recognizing the value of our work and for the constructive suggestions. Below, we carefully address each comment point by point.
>
> ## Q1: Extending domain generalization evaluation dimensions
> ## A1:
> Thank you for your suggestion. We would like to clarify that LIBERO-Gen already evaluates domain generalization along **five dimensions**, covering both perturbations that leave the expected action unchanged (e.g., *background*, *irrelevant objects*, and *language*) and those that require the action to change (e.g., *initial positions* and *task goals*). While not exhaustive, this already provides a structured basis for evaluating robustness.
>
> We agree that evaluating broader real-world factors is important for practical deployment, and we will consider extending LIBERO-Gen along these dimensions in future work. As a preliminary validation, we conduct a small-scale extension along two factors: lighting variation and sensor noise. Specifically, following the previous evaluation setup, we divide both illumination intensity and robot-body noise into five levels, construct corresponding training sets to fine-tune the Pi0.5 model, and analyze its performance under a consistent evaluation protocol (k1, k2, k3).
>
> **Table A. Evaluation of pi0.5 under lighting and sensor noise.**
> |Dimension|K1|K2|K3|
> |---|---:|---:|---:|
> |Lighting|0.985|1.000|0.995|
> |Sensor noise|0.980|0.970|0.970|
>
> These extension results are consistent with our observations in Table 1, showing that the model exhibits relatively strong generalization to input perturbations across vision, language, and robot-state noise.
>
> Due to time constraints, we leave further evaluation along additional dimensions to future work.
>
> ## Q2: Can show LIBERO-Gen correlates better with real-world than LIBERO?
> ## A2:
> Thank you for this important suggestion. LIBERO-Gen is designed to evaluate extrapolative generalization under controlled training support, which may be more relevant to deployment than standard in-distribution simulation scores.
>
> As a preliminary validation, we compared representative VLA models on standard LIBERO, LIBERO-Gen, and the real-robot RoboChallenge benchmark.
>
> **Table B. Comparison with real-robot benchmarks.**
> |Model|LIBERO (avg.,rank)|LIBERO-Gen (ours)|RoboChallenge|
> |---|---:|---:|---:|
> |π0.5|96.9,3|86.2,1|42.7,1|
> |π0|~96.0,4|79.7,2|28.3,2|
> |X-VLA|98.1,1|74.2,3|21.3,3|
> |OpenVLA|97.1,2|58.2,4|5.0,4|
>
> Although small-scale, this comparison provides a useful signal: **LIBERO-Gen aligns with the real-robot ranking, whereas standard LIBERO does not**.
>
> We will add this discussion to the revised manuscript and note that broader validation remains future work.
> ## Q3: Confidence intervals under Random vs. Stair sampling
> ## A3:
>
> Thank you for the suggestion. We would like to clarify that all results reported in the manuscript are means over 50 independent evaluation runs. To better quantify the reliability of the `Stair` vs. `Random` comparison, we report the sample variances and 95% confidence intervals in Table C and D.
>
> **Table C. Confidence intervals from repeated evaluations on position.**
>
> | Method         |  Mean | Sample Variance | 95% Confidence Interval |
> |----------------|------:|----------------:|-------------------------|
> | Pi0.5-stair    | 0.64 |       5.000e-3  | [0.607, 0.673]          |
> | Pi0.5-random   | 0.52 |       2.000e-3  | [0.499, 0.541]          |
> | OpenVLA-stair  | 0.26 |       1.286e-4  | [0.2547, 0.2653]        |
> | OpenVLA-random | 0.02 |       4.000e-7  | [0.0197, 0.0203]        |
>
> **Table D. Confidence intervals from repeated evaluations on task.**
> | Method         |  Mean | Sample Variance | 95% Confidence Interval |
> |----------------|------:|----------------:|-------------------------|
> | Pi0.5-stair    | 0.21 |       1.000e-3  | [0.195, 0.225]          |
> | Pi0.5-random   | 0.15 |       3.000e-3  | [0.124, 0.176]          |
> | OpenVLA-stair  | 0.00 |       0.000e+0  | [0.000, 0.000]          |
> | OpenVLA-random | 0.00 |       0.000e+0  | [0.000, 0.000]          |
>
> The improvement of `Stair` over `Random` is statistically significant for Pi0.5 in both tables (p = 3.24e-07 and p = 1.91e-04) and for OpenVLA in the first table (p = 5.37e-27). In the second table, both OpenVLA variants are identically zero across all 50 runs, so no significance test is reported.
>
> We will revise Fig. 8 accordingly to include this statistical information.

---

> > ### Author Rebuttal · Reviewer_3f33 · 2026-04-03
> >
> > Thank you for the detailed rebuttal. The authors have addressed my questions in a clear and helpful manner, and the clarifications have largely resolved my main concerns. Taking both the manuscript and the rebuttal into account, I believe that my current score continues to fairly reflect the overall quality of the paper, and I therefore plan to maintain my rating.

---

> > > ### Author Response · Authors · 2026-04-03
> > >
> > > We sincerely thank you again for your time and valuable suggestions. We greatly appreciate your recognition of our work and response.

---

### Official Review · Reviewer_ZYxT · 2026-03-06

**Soundness:** 2
**Presentation:** 3
**Significance:** 3
**Originality:** 2
**Overall Recommendation:** 4
**Confidence:** 4

**Summary:**

The paper argues that current Vision-Language-Action (VLA) evaluation protocols incentivize mechanical memorization over robust policy learning. It introduces LIBERO-Gen, a diagnostic benchmark that shifts evaluation toward explicit distributional assumptions across visual, linguistic, and spatial domains. By evaluating state-of-the-art models (e.g., Pi0.5, OpenVLA), the authors reveal severe performance collapses under minor out-of-distribution shifts. The study attributes these failures to two primary mechanisms: perceptual instability in the encoder and action binding collapse in the decoder.

**Compliance With Llm Reviewing Policy:**

Affirmed.

**Key Questions For Authors:**

1. Given the disparate pre-training data and backbones , can you provide a controlled ablation study isolating the action head to prove continuous flow-matching (Pi0.5) inherently resolves "Binding Collapse" over discrete modeling (OpenVLA)?"
2. Binding Collapse" Taxonomy: You define "Binding Collapse" as a decoder failure. However, if linear separability is already lost at the encoder's fused hidden states (e.g., 0.00% accuracy in Figure 7), isn't this fundamentally a representation failure?
3. Generalizability of "Stair" Sampling: The CG ($k=1$) evaluation explicitly tests the gaps created by the "Stair" training splits. Does this strategy improve true generalization on independent benchmarks, or does it merely overfit this specific evaluation protocol?

**Limitations:**

No.
1. Sim-to-Real Gap: It is unproven whether overcoming synthetic procedural shifts translates to physical robot reliability.
2. The danger of over-trusting generalist VLA models in unconstrained physical environments based solely on simulated scores.

**Strengths And Weaknesses:**

Strengths:
- The paper mathematically formalizes VLA policies within a POMDP and derives solid compositional generalization bounds based on causal inference. Linear probing effectively substantiates the claims regarding latent representation collapse.
- The theoretical decomposition of VLA failures into specific encoder-side (perceptual instability) and decoder-side (binding collapse) mechanisms provides a original diagnostic perspective.

Weaknesses:
- The paper defines "Binding Collapse" as a "Decoder Failure". However, the linear probing analysis reveals 0.00% accuracy for certain objects in the fused hidden states. If the encoder output loses linear separability of target objects, the failure fundamentally lies in the representation/encoder layer, contradicting its classification as a decoder failure.
- The benchmark relies solely on binary task success rates $Succ(\tau) \in \{0,1\}$. For a diagnostic suite, lacking fine-grained sub-metrics (e.g., grasp success rate, reaching accuracy) limits the ability to precisely pinpoint where the policy fails physically.
- In Section 3, the narrative frames the "Stair" and "L-shape" sampling strategies almost as novel methodological designs introduced by this paper. While the authors do cite relevant compositional generalization theories in Appendix A, there is no explicit attribution in the main text to the prior empirical literature that formalized these exact grid-based and diagonal holdout topologies for machine learning datasets. This must be corrected to properly reflect prior arts.
- While the benchmark elegantly exposes simulated generalization failures, it relies entirely on the simulated LIBERO environment. The paper claims that LIBERO-Gen establishes a "baseline for deployment reliability" , but without physical robot experiments, it remains unproven whether overcoming these specific simulated distributional shifts (e.g., procedural texture changes or LLM-paraphrased instructions ) directly translates to bridging the actual sim-to-real gap.

---

> ### Author Rebuttal · Authors · 2026-03-31
>
> We sincerely thank the reviewer for recognizing the value of our work and for the constructive suggestions. Below, we carefully address each comment point by point.
>
> ## Q1: Clarify whether "Binding Collapse" is a decoder or representation failure
> ## A1:
> We thank you for your suggestion. For out-of-distribution task failures, **decoder failure is sufficient but not necessary**. Such failures can arise from either:
> - **encoder-side failure**, where the encoder or fused representation fails to preserve separability of the target object;
> - **decoder-side failure**, where the decoder fails to bind actions under unseen compositions. For example, in **LIBERO-GOAL**, Pi0 can perform seen tasks such as `placing the bowl on the stove` and `placing the bowl on the plate`, but still fails on the unseen task `placing the plate on the stove`.
>
> We will revise the manuscript to clarify this point.
>
> ## Q2: Fine-grained evaluation metrics
> ## A2:
> We thank you for your suggestion. We use **binary task success** to follow standard practice in robotic manipulation benchmarks such as **LIBERO**, and because it is sufficient for the main goal of LIBERO-Gen: evaluating **generalization** under controlled training support across **ID/CG/DG**.
>
> At the same time, we agree that finer-grained metrics such as **grasp success**, **reaching accuracy**, and **object recognition success** would lead to more precise failure attribution and further strengthen the diagnostic value of the benchmark. We will clarify this limitation in the revision and highlight such metrics as an important direction for future work.
>
> ## Q3: Explicitly cite the origin of the Stair/L-shape design in the main text
> ## A3:
> Our main contribution is **not** these sampling topologies themselves, but the **hierarchical evaluation of VLA generalization under explicitly controlled training support**, i.e., the ID / CG / DG framework.
>
> We will revise the wording in the revised manuscript to better highlight the connection to prior techniques and more accurately reflect the existing techniques.
>
> ## Q4: Clarifying reliability in real-world deployment
> ## A4:
> LIBERO-Gen is designed to evaluate extrapolative generalization under controlled training support, which may be more relevant to deployment than standard in-distribution simulation scores.
>
> As a small-scale external check, we compared representative VLA models on standard LIBERO, LIBERO-Gen, and the real-robot RoboChallenge benchmark. **LIBERO-Gen produces a model ranking consistent with RoboChallenge, whereas standard LIBERO does not**.
>
> |Model|LIBERO(avg.,rank)|LIBERO-Gen (Ours)|RoboChallenge|
> |---|---:|---:|---:|
> |π0.5|96.9,3|86.2,1|42.7,1|
> |π0|~96.0,4|79.7,2|28.3,2|
> |X-VLA|98.1,1|74.2,3|21.3,3|
> |OpenVLA|97.1,2|58.2,4|5.0,4|
>
> We will clarify in the revised manuscript that broader real-world validation remains future work.
> ## Q5: Controlled action-head ablation for Binding Collapse
> ## A5:
> To address this concern, we conducted a controlled comparison using Pi0-fast, which shares the same backbone and pre-training data as Pi0 but adopts a discretized action output similar to OpenVLA. Specifically, we trained Pi0 and Pi0-fast on the same dataset, and the evaluation results are summarized in the table below.
>
> |Model|Position(ID/CG/DG)|Task|
> |---|---|---|
> |Pi0*|95.4/13.2/4.2|98.8/13.6/36.4|
> |Pi0-fast*|91.5/4.0/1.5|95.5/8.5/26.5|
>
> The results in the table show that Pi0 consistently outperforms Pi0-fast, and this controlled comparison further supports our claim.
>
> ## Q6: Does Stair sampling improve true generalization or merely fit the evaluation protocol?
> ## A6:
> “Stair” sampling is not intended as a universally better training strategy. Instead, its role in LIBERO-Gen is to provide a **controlled training support** for isolating **compositional generalization (CG)** from broader distribution shifts.
>
> Under the same data budget, more systematic factor coverage can better support **CG** than random sampling, but this does not imply gains for all forms of generalization. This is exactly why LIBERO-Gen separates **CG** from **DG**, rather than treating all unseen cases as a single type of generalization.
>
> We will revise the manuscript to make clear that “Stair” sampling mainly benefits compositional generalization (CG), rather than generalization in all settings.
>
> ## Q7: Add limitation discussion
> ## A7:
>
> We will add a dedicated discussion in the revision.
>
> Regarding the **sim-to-real gap**, we believe LIBERO-Gen can offer **directional evidence** for deployment, since real-world systems also require generalization under limited training data. However, it cannot replace real-robot validation, as deployment depends on many additional factors, including environmental complexity, embodiment differences, and hardware characteristics.
>
> Accordingly, LIBERO-Gen is intended to provide **controlled diagnostic insights and comparative guidance**, rather than to guarantee reliable performance in real-world deployment.

---

> > ### Author Rebuttal · Reviewer_ZYxT · 2026-04-02
> >
> > Thank you for the detailed rebuttal. All my concerns have been adequately addressed by the authors' response. I plan to maintain my current score, as I believe it remains a fair and accurate reflection of the paper's overall quality following these clarifications.

---

> > > ### Author Response · Authors · 2026-04-02
> > >
> > > We sincerely thank you for your time and insightful suggestions. We also greatly appreciate your recognition of our work and response.

---

### Official Review · Reviewer_1oTU · 2026-03-12

**Soundness:** 3
**Presentation:** 3
**Significance:** 3
**Originality:** 2
**Overall Recommendation:** 5
**Confidence:** 3

**Summary:**

The paper introduces a benchmark named LIBERO-Gen that systematically studies different generalization aspects of VLA models. In particular, the paper studies three evaluation regimes—in-domain, compositional, and domain generalization—across several variation factors, including background, language, distractor objects, task semantics, and spatial configuration, to expose reliance on spurious correlations rather than true robustness. The paper also presents an analysis of failure mechanisms and attempts to mitigate them through augmentation, structured data sampling, and modified action representations. Another aspect of the work is an attempt to separate perceptual failures from action-level binding failures.

**Compliance With Llm Reviewing Policy:**

Affirmed.

**Final Justification:**

My concerns about data contamination and paper positioning have been addressed.

**Key Questions For Authors:**

Q: “Models fine-tuned on LIBERO-Gen specific subsets.” Does this mean that the model was fine-tuned on the in-domain part, or each subset (e.g., k3, DG)?

Q: When evaluating models' generalization, it is important to avoid data contamination. Can you provide some evidence that the models haven't had the data during pre-training?

Q: Please, address the novelty issue.

**Limitations:**

yes

**Strengths And Weaknesses:**

Strength: The paper addresses a genuinely important robotics problem, since real-world robot deployment depends on robustness to distribution shift rather than success on narrow benchmark settings.

Strength: It proposes a clear and relatively rigorous generalization framework by separating in-domain, compositional, and domain generalization across controlled variation factors.

Strength: The failure analysis is useful because it goes beyond reporting performance drops and identifies more specific mechanisms behind model failures.

Strength: The paper makes the analysis more actionable by testing targeted interventions, even if they only partially address the problem

Weakness: the paper does not position itself clearly enough relative to prior benchmark work on perturbation-based and out-of-distribution evaluation, especially The Colosseum and AGNOSTOS [1,2]. This matters because the core idea of systematically probing robustness under controlled variation is not entirely new: The Colosseum already studies manipulation robustness under a broad set of environmental perturbations, such as changes in color, texture, size, lighting, distractors, camera pose, and physical properties, while AGNOSTOS evaluates zero-shot cross-task generalization on unseen manipulation tasks. A more explicit comparison would help clarify what is genuinely new here—whether it is the specific decomposition into in-domain / compositional / domain generalization, the choice of variation axes, or the accompanying failure analysis—rather than leaving the impression that this benchmark space is less developed than it already is. Related recent benchmarks such as LIBERO-Plus and INT-ACT make this omission more noticeable, since they likewise analyze VLA brittleness under controlled perturbations and targeted generalization probes [3,4].

[1]. arXiv:2402.08191v2

[2]. arXiv:2505.15660v3

[3]. arXiv:2510.13626v3

[4]. arXiv:2506.09930v1

---

> ### Author Rebuttal · Authors · 2026-03-31
>
> We sincerely thank the reviewer for recognizing the value of our work and for the constructive suggestions. Below, we carefully address each comment point by point.
>
> ## Q1: Clarify the distinct from prior robustness/OOD benchmarks
>
> ## A1:
> Thank you for this helpful suggestion.
>
> Unlike prior work, which often treats any training-unseen input as generalization, **we define VLA generalization under explicit data distribution assumptions**. Under this framework, we distinguish **compositional generalization (CG)** from **domain generalization (DG)**, enabling controlled evaluation of model behavior under different shifts.
>
> **Table A. Comparison with prior work.**
> |Benchmark|Focus|OOD notion|
> |---|---|---|
> |LIBERO-Plus|Perturbation robustness|Heuristic perturbations|
> |THE COLOSSEUM|Manipulation robustness|Environmental/physical changes|
> |AGNOSTOS|Cross-task generalization|Unseen tasks|
> |INT-ACT|Intention vs. execution|Unseen objects/language|
> |**LIBERO-Gen**|**Generalization under controlled distributions**|**ID/CG/DG**|
>
> Our results further show **a clear gap between CG and DG**, indicating that VLA generalization depends not just on whether a task is “seen,” but on the support of the training distribution and the type of shift. This also helps explain failures under different generalization regimes.
>
> We will revise the paper to make this distinction clearer.
>
> ## Q2: Address the novelty issue
>
> ## A2:
> **Novelty of our work.**
>
> - **Motivation.** We argue that VLA generalization should not be characterized only by an intuitive **seen/unseen** split. A key limitation of this view is that it lacks a precise **distributional foundation**, making it difficult to distinguish different sources of generalization failure.
>
> - **Method.** We introduce a **strict data-distribution-based formulation** of VLA generalization. By defining VLA tasks with respect to the underlying data distribution, we formally distinguish **ID**, **CG**, and **DG**, yielding a more precise and rigorous framework than heuristic seen/unseen definitions.
>
> - **Findings.** Our experiments reveal substantial performance differences between **CG** and **DG**, showing that the key factor in VLA generalization is the **distribution structure of the data**, rather than simple task novelty. In addition, by **decoupling the VLA encoder and decoder**, we conduct a detailed failure analysis that offers a new diagnostic perspective on why VLA models fail to generalize, and suggests concrete directions for improvement.
>
> ## Q3: Clarify whether fine-tuning uses only the in-domain split or each specific subset (e.g., k3, DG)
>
> ## A3:
> Thank you for pointing this out. **The fine-tuned models were trained only on the suite-specific training split, i.e., $D_{\text{train}}$, and not on evaluation subsets such as CG or DG.**
>
> Concretely, we construct $D_{\text{train}}$ using the “Stair” and “L”-shaped sampling schemes, and evaluate:
> - `ID (k=0)`: **seen factors, seen compositions**;
> - `CG (k=1)`: **seen factors, unseen compositions**;
> - `DG (k=2)`: **unseen factors, unseen compositions**.
>
> We will clarify this explicitly in the revised manuscript.
>
> ## Q4: Provide evidence against data contamination from pre-training during generalization evaluation
>
> ## A4:
> Thank you for raising this important concern.
>
> **LIBERO-Gen is not a simple reuse of existing LIBERO instances.** Its evaluation set is newly constructed under **controlled distribution shifts**, including **novel combinations of task-relevant factors** and **novel combinations of additional novel factors** beyond the original training distribution.
>
> More importantly, VLA policies are learned over the **joint distribution of vision, language, and robot states/actions** rather than any single modality alone. Thus, the appearance of an individual texture, object, or language expression in pre-training does not by itself imply contamination.
>
> For models such as **pi0**, we cannot fully audit all pre-training data. However, we provide **indirect evidence** by evaluating the **pretrained pi0** and **pi0.5** checkpoints in a zero-shot setting on our benchmark.
>
> **Table B. Pretrained model performance on LIBERO-Gen.**
>
> |Model|Bg.(k1/k2/k3)|Lang.|Obj.|Pos.|Task|
> |---|---:|---:|---:|---:|---:|
> |pi0|0.0/0.0/0.0|0.0/0.0/0.0|0.0/0.0/0.0|0.0/0.0/0.0|0.0/0.0/0.0|
> |pi0.5|0.0/0.0/0.0|0.0/0.0/0.0|0.0/0.0/0.0|0.0/0.0/0.0|0.0/0.0/0.0|
>
> As shown in **Table B**, the pretrained models perform very poorly, suggesting that contamination is unlikely to explain the observed gap.
>
> Moreover, even with possible exposure to some individual benchmark components during pre-training, the models still fail under our controlled recompositions and perturbations. This supports our central claim that **familiarity with benchmark elements does not imply robust extrapolative generalization under structured distribution shift**.

---

> > ### Author Rebuttal · Reviewer_1oTU · 2026-04-03
> >
> > Thank you for the clarifications. My concerns have been fully resolved, and therefore, I am raising my score. Please include the comparison with the related works in the paper revision.

---

> > > ### Author Response · Authors · 2026-04-03
> > >
> > > We sincerely thank you for your time and insightful suggestions. We will ensure that the comparison with related works is included in the revised manuscript. We also greatly appreciate your recognition of our work and response.

---

### Official Review · Reviewer_E1k4 · 2026-03-13

**Soundness:** 2
**Presentation:** 3
**Significance:** 3
**Originality:** 3
**Overall Recommendation:** 4
**Confidence:** 3

**Summary:**

This paper addresses the sim2real gap by developing LIBERO-Gen, a benchmark that evaluates how well Vision-Language-Action models generalize across distributional shifts in simulation. To do so, the authors introduce new semantic compositions and environmental changes as evaluation conditions and measure whether these models have learned robust policies rather than relying on mechanical memorization. They show many high-scoring models on standard benchmarks exhibit spurious invariance to semantics and brittleness to small perturbations. The results show that spatial and task generalization are dominant failure modes while structured sampling and encoder regularization can partially mitigate failures.

**Compliance With Llm Reviewing Policy:**

Affirmed.

**Key Questions For Authors:**

1. How many rollouts per task and random seeds were used to produce the main results in Tables 1 and 2? And can you provide per-axis confidence intervals or standard errors? Without any measure of variance it is difficult to judge whether differences across models or across training strategies are statistically meaningful or within noise.

2. In the domain-generalization regime for language, are paraphrases a new domain, or would that be better considered compositional or semantic variation? Could you please formalize domain boundary definitions per axis?

3. Have you considered a small real-robot or sim–real correlation study for a subset of axes to validate that LIBERO-Gen scores predict real-world robustness? I need to mention that not doing so wouldn’t invalidate the contribution, but would clarify the scope of the claims.

**Limitations:**

yes

**Strengths And Weaknesses:**

Strengths:
- The paper studies a wide range of generalizations dimension (background, language, distractors, task semantics, spatial configuration) with 3 levels of distribution shift (in-distribution, compositional generalization, domain generalization).
- The benchmark design and hierarchical protocol are clearly described. Formalization of compositional and domain generalization helps conceptualizing the evaluation protocol.
- It addresses a critical bottleneck in spatial extrapolation and semantic binding in VLA models, which makes it valuable for the community.
- The explicit distributional control and failure-mode analysis are insightful and novel.
---
Weaknesses:
- Lack of sufficient statistical reporting like the number of rollouts per task, confidence intervals, and etc. make it hard to judge the results.
- Some key design choices like why exactly 5 groups or why sliding windows of size 2 lack explicit justification.
- Context related to domain randomization works like RoboTwin could be more explicit.
- The domain generalization axis leans heavily on texture and material changes. For language domain generaliztion, it is not clear that paraphrases constitute a distinct domain.

---

> ### Author Rebuttal · Authors · 2026-03-31
>
> We sincerely thank the reviewer for recognizing the value of our work and for the constructive suggestions. Below, we carefully address each comment point by point.
>
> ## Q1: Reporting Result Statistics
>
> ## A1:
> **The main results in Tables 1 and 2 are all reported as means over 50 independent evaluation runs.**
>
> Due to space limitations, we provide here the 95% confidence intervals for the pi0 and pi0.5 results in Table 1, as representative example.
> |G|i|pi0|pi0.5|
> |---|--:|---|---|
> |position|0|0.95±0.03|0.98±0.03|
> ||1|0.07±0.02|0.62±0.04|
> ||2|0.05±0.02|0.25±0.05|
> |task|0|0.98±0.02|0.99±0.01|
> ||1|0.14±0.03|0.20±0.03|
> ||2|0.40±0.05|0.52±0.06|
> |object|0|0.91±0.03|0.99±0.02|
> ||1|0.98±0.02|1.00±0.00|
> ||2|0.88±0.04|1.00±0.01|
> |texture|0|0.97±0.02|1.00±0.01|
> ||1|0.96±0.02|1.00±0.00|
> ||2|0.99±0.02|1.00±0.00|
> |language|0|0.99±0.02|1.00±0.01|
> ||1|0.98±0.02|1.00±0.01|
> ||2|0.95±0.04|0.99±0.01|
>
> Each entry is reported as mean ± half-width of the 95% confidence interval. These intervals are consistent with and support the conclusions drawn in the paper.
>
> **In the revised manuscript, we will include confidence intervals and variance statistics for all experiments.**
>
> ## Q2: Insufficient justification for key design choices
> ## A2:
> We use a window size of 2 because it is the most reasonable choice in our compositional-generalization setup, and is also consistent with prior experience [1][2] from dataset construction:
>
> - if the window size were 1, some context values would be tied to only one task, encouraging shortcut memorization over transferable compositional learning;
> - if the window size were larger than 2, the training support would become much denser, weakening the intended sparse compositional challenge.
>
> Given this size-2 design, and since these suites are built on the 10 base tasks inherited from LIBERO, the grouping is naturally induced as 5 groups rather than being chosen independently.
>
> Ref:\
> [1] Compositional generalization from first principles\
> [2] Efficient data collection for robotic manipulation via compositional generalization
>
> ## Q3: Need clearer positioning against domain randomization work (e.g., RoboTwin)
>
> ## A3:
>  We will revise the related-work section to better clarify LIBERO-Gen’s connection to and distinction from prior domain-randomization-related works.
>
> Prior domain-randomization-related works typically assess generalization through broad “unseen” perturbations or heuristic benchmark design. In contrast, **LIBERO-Gen is grounded in a rigorous theoretical framework that defines generalization at three levels: ID (in-distribution generalization), CG (unseen combinations of seen factors), and DG (out-of-support shifts)**.
>
> Precisely because of this distinction, our results show that CG and DG should not be treated uniformly: CG can be improved by more balanced and structured sampling, whereas DG remains substantially more challenging.
>
>
> ## Q4: Unclear domain boundaries in language DG
>
> ## A4:
> In VLA, actions are learned from the **joint embodied distribution** over vision, language, and robot state, rather than from language alone. Thus, whether an instruction is in-distribution depends not only on semantic equivalence, but on whether it lies within the joint task–context support seen during training.
>
> In our benchmark, language-ID and language-CG use **seen template families**, while language-DG uses **out-of-template instructions**, including GPT-generated variants with different wording, syntax, and contextual framing. Because these instructions never co-occur with the corresponding visual and robot-state context during training, we categorize them as DG. We will revise the paper to make this distinction explicit.
>
> More generally, we will clarify the DG boundary for each axis as follows:
> - **Background/Texture:** unseen texture or material families.
> - **Distractor Object:** unseen distractor-object pools or configurations.
> - **Language:** unseen out-of-template instructions, including GPT-generated paraphrases and noisy/contextual variants.
> - **Spatial Configuration:** object or target positions outside the seen discrete grid support.
> - **Task Semantics:** unseen paired-object task structures beyond the sparse composite-task support observed during training.
>
> ## Q5: Have you considered conducting real-robot experiments?
>
> ## A5:
> As a small-scale external check, we compared representative VLA models on standard LIBERO, LIBERO-Gen, and the real-robot RoboChallenge benchmark. **LIBERO-Gen yields a ranking consistent with RoboChallenge, whereas standard LIBERO does not**, suggesting that it may better capture deployment-relevant differences across models.
>
> |Model|LIBERO(avg.,rank)|LIBERO-Gen|RoboChallenge|
> |---|---:|---:|---:|
> |π0.5|96.9,3|86.2,1|42.7,1|
> |π0|~96.0,4|79.7,2|28.3,2|
> |X-VLA|98.1,1|74.2,3|21.3,3|
> |OpenVLA|97.1,2|58.2,4|5.0,4|
>
> We will further evaluate this in future work through targeted real-robot studies on a subset of axes.

---

> > ### Author Rebuttal · Reviewer_E1k4 · 2026-04-04
> >
> > Thank you for the clarifications. Your response addresses my concerns sufficiently, and I think they strengthen the paper. However, I think my current rating fairly reflects the paper's standing given the clarifications.

---

> > > ### Author Response · Authors · 2026-04-04
> > >
> > > Thank you very much for your thoughtful follow-up. We are deeply grateful for your positive feedback and sincerely appreciate your recognition of our work and response.

---

### Decision · Program_Chairs · 2026-04-30

**Decision:**

Accept (regular)

**Comment:**

This paper introduces LIBERO-Gen, an evaluation benchmark for Vision-Language-Action (VLA) models designed to reveal generalization limits through explicit distributional shifts. The research finds that many models with high scores on standard tests actually rely on "mechanical memorization," leading to perceptual instability and action binding collapse when faced with minor environmental changes or semantic perturbations. The authors demonstrate that models using continuous action modeling, such as Pi0.5, combined with structured sampling strategies, significantly improve resilience to these shifts.

The paper’s strengths lie in its in-depth failure mode analysis, which successfully decouples the distinct roles of the encoder and decoder in generalization failures. Importantly, the authors provided evidence during the rebuttal that LIBERO-Gen rankings correlate strongly with physical robot performance (RoboChallenge), proving it is a more reliable indicator than traditional benchmarks. While reviewers initially questioned the statistical completeness (e.g., confidence intervals) and the comparison with existing works like The Colosseum, the authors addressed these points professionally and proactively.

In rebuttal, the authors supplemented the 95% confidence intervals and detailed the differences from prior works, the core concerns were resolved, leading to a positive final evaluation from reviewers. Given the paper’s practical contribution to the robotics community, it is decided to be accepted.